# Dual Cone Gradient Descent for Training Physics-Informed Neural Networks

**Youngsik Hwang**
Artificial Intelligence Graduate School
UNIST
hys3835@unist.ac.kr

**Dong-Young Lim**[*]
Department of Industrial Engineering
Artificial Intelligence Graduate School
UNIST
dlim@unist.ac.kr

## Abstract

Physics-informed neural networks (PINNs) have emerged as a prominent approach for solving partial differential equations (PDEs) by minimizing a combined loss function that incorporates both boundary loss and PDE residual loss. Despite their remarkable empirical performance in various scientific computing tasks, PINNs often fail to generate reasonable solutions, and such pathological behaviors remain difficult to explain and resolve. In this paper, we identify that PINNs can be adversely trained when gradients of each loss function exhibit a significant imbalance in their magnitudes and present a negative inner product value. To address these issues, we propose a novel framework for multi-objective optimization, *Dual Cone Gradient Descent* (DCGD), which adjusts the direction of the updated gradient to ensure it falls within a dual cone region. This region is defined as a set of vectors where the inner products with both the gradients of the PDE residual loss and the boundary loss are non-negative. Theoretically, we analyze the convergence properties of DCGD algorithms in a non-convex setting. On a variety of benchmark equations, we demonstrate that DCGD outperforms other optimization algorithms in terms of various evaluation metrics. In particular, DCGD achieves superior predictive accuracy and enhances the stability of training for failure modes of PINNs and complex PDEs, compared to existing optimally tuned models. Moreover, DCGD can be further improved by combining it with popular strategies for PINNs, including learning rate annealing and the Neural Tangent Kernel (NTK). Codes are available at https://github.com/youngsikhwang/Dual-Cone-Gradient-Descent.

## 1 Introduction

Physics-informed Neural Networks (PINNs) proposed in Raissi et al. [1] have created a new paradigm in deep learning for solving forward and inverse problems involving partial differential equations (PDEs). The key idea of PINNs is to integrate physical constraints, governed by PDEs, into the loss function of neural networks. This is in turn equivalent to finding optimal parameters for the neural network by minimizing a loss function that combines boundary loss and PDE residual loss. Thanks to their strong approximation ability and mesh-free advantage, PINNs have achieved great success in a wide range of applications [2–8].

Building upon this success, the applications of PINNs have been extended to solve other functional equations, including integro-differential equations [9], fractional PDEs [10], and stochastic PDEs [11]. Moreover, numerous variants of PINNs have been developed to enhance their computational efficiency and accuracy via domain decomposition methods [12, 13], advanced neural network architectures [14–18], modified loss functions [19–21], different sampling strategies [22–24], and

---

[*]Corresponding author.

38th Conference on Neural Information Processing Systems (NeurIPS 2024).

probabilistic PINNs [25, 26]. Recent studies have also explored optimizing PINNs by leveraging function space geometry, providing an alternative perspective to enhance accuracy and computational efficiency [27, 28].

Despite these achievements, several studies have reported that PINNs often fail to learn correct solutions for given problems ranging from highly complex to relatively simple PDEs [29–31]. Due to the unclear nature of pathologies in the training of PINNs, it has become a critical research topic to explain and mitigate these phenomena. For example, [32, 33] observed that PINNs tends to get stuck at trivial solutions while violating given PDE constraints over collocation points. The imbalance between PDE residual loss and boundary loss was explored in Wang et al. [30], and a spectral bias of PINNs was studied in Wang et al. [31]. Yao et al. [34] discussed the gap between the loss function and the actual performance. Even with the insights from the aforementioned studies, a comprehensive understanding of PINN's failure modes remains largely unexplored in various scenarios.

In this paper, we explore these challenges from a novel perspective of multi-objective optimization. We first provide a geometric analysis showing that PINNs can be adversely trained when the gradients of each loss function exhibit a significant imbalance in their magnitudes, coupled with a negative inner product value. Based on this finding, we characterize a dual cone region where both PDE residual loss and boundary loss can be decreased simultaneously without the adverse training phenomenon. We then propose a novel optimization framework, *Dual Cone Gradient Descent* (DCGD), for training PINNs which updates the gradient direction to be contained in the dual cone region at each iteration. Furthermore, we study the convergence properties of DCGD in a non-convex setting. In particular, we find that DCGD can converge to a Pareto-stationary point. We validate the superior empirical performance and universal applicability of DCGD through extensive experiments.

## 2 Preliminaries

**Notation.** The Euclidean scalar product is denoted by $\langle \cdot, \cdot \rangle$, with $\| \cdot \|$ standing for the Euclidean norm (where the dimension of the space may vary depending on the context). For a subspace $W$ of a vector space $V$, its orthogonal complement $W^\perp$ is defined as

$$W^\perp := \{v \in V | \langle u, v \rangle = 0, \quad u \in W\}.$$

For a vector $v \in V$, the projection of $v$ on a nontrivial subspace $W$ is denoted by $v_{\|W}$. Unless otherwise specified, $V$ represents $\mathbb{R}^d$ throughout the paper.

**Related Works.** Among various research directions in PINNs, we focus on reviewing optimization strategies for PINNs. These can be broadly categorized into three main approaches: adaptive loss balancing, gradient manipulation, and Multi-Task Learning (MTL). As an example of adaptive loss balancing algorithms, Wang et al. [30] proposed a learning rate annealing (LRA) algorithm that balances the loss terms by utilizing gradient statistics. Wang et al. [31] utilized the eigenvalues of the Neural Tangent Kernel (NTK) to address the disparity in convergence rates among different losses of PINNs. For gradient manipulation algorithms, the Dynamic Pulling Method (DPM) was proposed in [35] to prioritize the reduction of the PDE residual loss. In [36], the authors used the PCGrad algorithm, proposed in [37], for training PINNs to address multi-task learning challenges. Li et al. [38] developed an adaptive gradient descent algorithm (AGDA) that resolves the conflict by projecting boundary condition loss gradient to the normal plane of the PDE residual loss gradient. Yao et al. [34] recently developed MultiAdam, a scale-invariant optimizer, to mitigate the domain scaling effect in PINNs. Another important line of gradient manipulation involves Multi-Task Learning (MTL) algorithms, which optimize a single model to perform multiple tasks simultaneously [39, 40, 37, 41–44]. We will discuss that several MTL algorithms can be unified within the proposed DCGD framework.

**Physics-Informed Neural Networks.** Let $\Omega \subseteq \mathbb{R}^D$ be a domain and $\partial\Omega$ be the boundary of $\Omega$. We consider the following nonlinear PDEs:

$$\begin{aligned} \mathcal{N}[u](\boldsymbol{x}) &= f(\boldsymbol{x}), \quad \boldsymbol{x} \in \Omega \\ \mathcal{B}[u](\boldsymbol{x}) &= g(\boldsymbol{x}), \quad \boldsymbol{x} \in \partial\Omega \end{aligned} \qquad (2.1)$$

where $\mathcal{N}$ and $\mathcal{B}$ denote a nonlinear differential operator and a boundary condition operator, respectively. We approximate $u(\boldsymbol{x})$ by a deep neural network $u(\boldsymbol{x}; \theta)$ parameterized by $\theta$. To train the

neural network, the framework of PINNs minimizes the total loss function $\mathcal{L}(\theta)$, which is a weighted sum of PDE residual loss $\mathcal{L}_r(\theta)$ and boundary condition loss $\mathcal{L}_b(\theta)$, defined by:

$$\mathcal{L}(\theta) := \omega_r \mathcal{L}_r(\theta) + \omega_b \mathcal{L}_b(\theta) \quad \text{with} \tag{2.2}$$

$$\mathcal{L}_r(\theta) := \frac{1}{N_r} \sum_{i=1}^{N_r} |\mathcal{N}[u(\cdot;\theta)](\boldsymbol{x}_r^i) - f(\boldsymbol{x}_r^i)|^2, \quad \mathcal{L}_b(\theta) := \frac{1}{N_b} \sum_{i=1}^{N_b} |\mathcal{B}[u(\cdot;\theta)](\boldsymbol{x}_b^i) - g(\boldsymbol{x}_b^i)|^2,$$

where $\omega_r, \omega_b \geq 0$ are weights of each loss term, $\{\boldsymbol{x}_r^i\}_{i=1}^{N_r}$ denotes a set of collocation points that are randomly sampled in $\Omega$, and $\{\boldsymbol{x}_b^i\}_{i=1}^{N_b}$ the boundary sample points. Here, we set $\omega_r = \omega_b = 1$ throughout the paper. We note that the training of PINNs falls into the category of multi-objective learning due to its structure of the loss function $\mathcal{L}(\theta)$ in Eq. (2.2).

# 3 Empirical Observations and Issues in Training PINNs

This section investigates issues that are frequently observed during the training of PINNs in the context of multi-objective learning. The parameter for the PINN solution $u(\boldsymbol{x};\theta)$ is typically estimated by minimizing the total loss function $\mathcal{L}(\theta)$ with a (stochastic) gradient descent method[2]:

$$\theta_{t+1} = \theta_t - \lambda \nabla \mathcal{L}(\theta_t), \quad t \in \mathbb{N}$$

where $\nabla \mathcal{L}(\theta)$ is the gradient of the total loss function $\mathcal{L}(\theta)$ with respect to $\theta$. However, a careless adoption of standard gradient descent methods may lead to an incorrect solution, as reducing the total loss does not necessarily

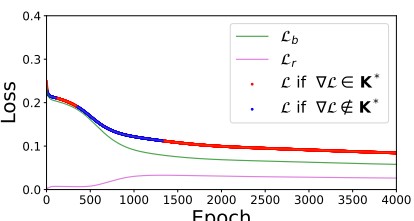

Figure 1: Training curves for the total loss $\mathcal{L}$ ($:= \mathcal{L}_r + \mathcal{L}_b$), PDE residual loss $\mathcal{L}_r$, and boundary loss $\mathcal{L}_b$ for viscous Burgers' equation.

imply a decrease in both the PDE residual loss and boundary loss. This phenomenon is clearly illustrated in Figure 1, which displays the curves of the total loss, PDE residual loss, and boundary loss over epochs for solving the viscous Burger's equation. Notably, while the total loss consistently decreases throughout the training, the PDE loss adversely increases.

**Conflicting and dominating gradients in PINNs.** This issue is highly related with discrepancies in the direction and magnitude between two gradients of the PDE residual and boundary loss. Specifically, we define two gradients to be *conflicting* at the $t$-th iteration if they have a negative inner product value, i.e., $\frac{\pi}{2} < \phi_t \leq \pi$ where $\phi_t$ is the angle between $\nabla \mathcal{L}_r(\theta_t)$ and $\nabla \mathcal{L}_b(\theta_t)$. When there are conflicting gradients, parameter updates to minimize one loss function might increase the other, leading to an inefficient learning process such as oscillating between optimizing for the two loss functions and resulting in degraded solution quality [19]. Another problem arises when one gradient is much larger than the other, i.e., $\|\nabla \mathcal{L}_r(\theta_t)\| \ll \|\nabla \mathcal{L}_b(\theta_t)\|$ or $\|\nabla \mathcal{L}_r(\theta_t)\| \gg \|\nabla \mathcal{L}_b(\theta_t)\|$. The significant differences[3] in the magnitudes of gradients in PINNs might create a situation where the optimization algorithm primarily minimizes one loss function while neglecting the other. This often results in slow convergence and overshooting, as the smaller gradient, though neglected, may be more crucial in finding a better solution. To mitigate the imbalance in the gradients, loss balancing approaches to rescale the weights of each loss term have been proposed [30, 31].

To examine these challenges in training PINNs, we record cosine value of the angle between $\nabla \mathcal{L}_r$ and $\nabla \mathcal{L}_b$, and the ratios of their magnitudes while training a PINN for the Helmholtz equation. Figure 2(a) shows that conflicting gradients are observed in about half of the total iterations. Moreover, we observe that the magnitude of the gradient of the PDE residual is several tens to hundreds of times larger than that of the boundary loss (See Figure 2(b)). That is, conflicting and dominating gradients are prevalent issues in the training of PINNs.

---

[2]In practice, adaptive gradient descent algorithms such as ADAM [45] are widely employed.

[3]Our subsequent analysis in Section 4.1 will clearly identify the extent to which significant differences in gradient magnitude lead to a challenge.

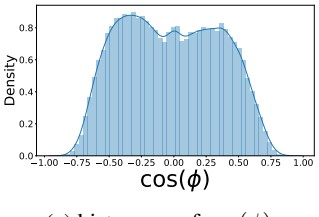

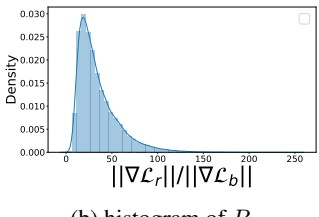

(a) histogram of $\cos(\phi)$       (b) histogram of $R$

Figure 2: Conflicting and dominating gradients in PINNs. Here, $\phi$ is defined as the angle between $\nabla\mathcal{L}_r$ and $\nabla\mathcal{L}_b$, $R = \frac{\|\nabla\mathcal{L}_r\|}{\|\nabla\mathcal{L}_b\|}$ is the magnitude ratio between gradients.

# 4 Methodology

In this section, we provide a geometric analysis to identify a dual cone region where both the PDE residual loss and the boundary loss can decrease simultaneously. Subsequently, we introduce a general framework for DCGD algorithms, ensuring that the updated gradient falls within this region. We then propose three specifications of DCGD algorithms: projection, average, and center. All proofs for main results in this section can be found in Appendix A.

## 4.1 Dual Cone Region

The concept of a dual cone plays a pivotal role in our DCGD algorithm. Formally, a dual cone is defined as a set of vectors that have nonnegative inner product values with a given cone.

**Definition 4.1.** (Dual cone) Let $\mathbf{K}$ be a cone of $\mathbb{R}^d$. Then, the set

$$\mathbf{K}^* = \{y | \langle x, y \rangle \geq 0 \quad \text{for all } x \in \mathbf{K}, y \in \mathbb{R}^d\}$$

is called the *dual cone* of $\mathbf{K}$.

For each iteration $t$, consider a cone denoted by $\mathbf{K}_t$, which is generated by rays of two gradients, $\nabla\mathcal{L}_r(\theta_t)$ and $\nabla\mathcal{L}_b(\theta_t)$:

$$\mathbf{K}_t := \{cx | c \geq 0, x \in \{\nabla\mathcal{L}_r(\theta_t), \nabla\mathcal{L}_b(\theta_t)\}\}.$$

In the context of PINNs, the dual cone of $\mathbf{K}_t$, denoted by $\mathbf{K}_t^*$, represents the set of gradient vectors where each vector is neither conflicting with the gradient of the PDE loss nor with the gradient of the boundary loss, i.e., for $u \in \mathbf{K}_t^*$, $\langle u, \nabla\mathcal{L}_r(\theta_t) \rangle \geq 0$ and $\langle u, \nabla\mathcal{L}_b(\theta_t) \rangle \geq 0$.

In other words, when the total gradient $\nabla\mathcal{L}(\theta_t)$ is in $\mathbf{K}_t^*$ (as depicted by the region of the red line in Figure 1), the standard gradient descent taking the direction $\nabla\mathcal{L}(\theta_t)$ will decrease both the PDE and boundary losses for a suitable step size. On the other hand, if $\nabla\mathcal{L}(\theta_t) \notin \mathbf{K}_t^*$ (the region indicated by the blue line in Figure 1), one of the two losses will adversely increase even with sufficiently small step sizes.

This indicates that the training process of PINNs can significantly vary depending on whether the total gradient belongs to the dual cone region. The following theorem establishes the necessary and sufficient conditions under which the total gradient falls within the dual cone region in terms of the angle and relative magnitude between the gradients of the PDE residual and boundary loss.

**Theorem 4.2.** *Suppose that $\nabla\mathcal{L}_r(\theta_t)$ and $\nabla\mathcal{L}_b(\theta_t)$ are given at each iteration $t$. Let $\phi_t$ be the angle between $\nabla\mathcal{L}_r(\theta_t)$ and $\nabla\mathcal{L}_b(\theta_t)$, and $R = \frac{\|\nabla\mathcal{L}_r(\theta_t)\|}{\|\nabla\mathcal{L}_b(\theta_t)\|}$ be their relative magnitude. Then, $\nabla\mathcal{L}(\theta_t) \in \mathbf{K}_t^*$ if and only if*

$$(i)\ \langle \nabla\mathcal{L}_b(\theta_t), \nabla\mathcal{L}_r(\theta_t) \rangle \geq 0\,, \text{ or}$$

$$(ii)\ \langle \nabla\mathcal{L}_b(\theta_t), \nabla\mathcal{L}_r(\theta_t) \rangle < 0 \text{ and } -\cos\phi_t \leq R \leq -\frac{1}{\cos\phi_t}.$$

Theorem 4.2 provides a clear criterion for when conflicting and dominating gradients lead to adverse training in PINNs. For instance, the condition (ii) in Theorem 4.2 implies that the larger $\phi_t$ (the more conflicting they are), even a slight difference in their magnitudes can result in adverse training. In particular, Theorem 4.2 quantifies the extent of problematic relative magnitude between the two

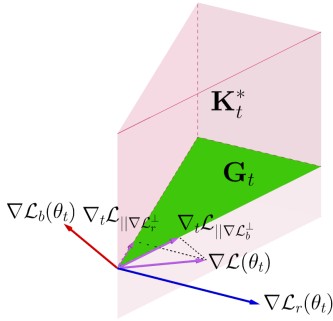

Figure 3: Visualization of dual cone region $\mathbf{K}_t^*$ and its subspace $\mathbf{G}_t$

gradients, thereby clarifying the concept of dominating gradients, which has not been previously defined in the literature.

Thus, our strategy aims to devise an algorithm that chooses the updated gradient within the dual cone region at each gradient descent step. For notational simplicity, we write $\nabla_t \mathcal{L}_{\|\nabla \mathcal{L}_r^\perp}$ and $\nabla_t \mathcal{L}_{\|\nabla \mathcal{L}_b^\perp}$ to represent $\nabla \mathcal{L}(\theta_t)_{\|(\nabla \mathcal{L}_r(\theta_t))^\perp}$ and $\nabla \mathcal{L}(\theta_t)_{\|(\nabla \mathcal{L}_b(\theta_t))^\perp}$, respectively. In particular, we are interested in a simple and explicit subspace $\mathbf{G}_t$, defined as the set of conic combinations of $\nabla_t \mathcal{L}_{\|\nabla \mathcal{L}_r^\perp}$ and $\nabla_t \mathcal{L}_{\|\nabla \mathcal{L}_b^\perp}$:

$$\mathbf{G}_t := \left\{ c_1 \nabla_t \mathcal{L}_{\|\nabla \mathcal{L}_r^\perp} + c_2 \nabla_t \mathcal{L}_{\|\nabla \mathcal{L}_b^\perp} \, \big| c_1, c_2 \geq 0 \right\}, \tag{4.1}$$

for two reasons. Firstly, all vectors in $\mathbf{G}_t$ are easily computable due to the explicit expression of $\mathbf{G}_t$, whereas the dual cone $\mathbf{K}^*$ is implicitly defined. Secondly, $\mathbf{G}_t$ contains two important components of $\mathbf{K}_t^*$, which are the projections of $\nabla \mathcal{L}(\theta_t)$ onto $\nabla \mathcal{L}_r(\theta_t)^\perp$ and $\nabla \mathcal{L}_b(\theta_t)^\perp$ by its construction. The next proposition shows that $\mathbf{G}_t$ always belongs to the dual cone region as illustrated in Figure 3.

**Proposition 4.3.** *Suppose that $\nabla \mathcal{L}_r(\theta_t)$ and $\nabla \mathcal{L}_b(\theta_t)$ are given at each iteration $t$. Consider $\mathbf{G}_t$, the set of conic combinations of $\nabla_t \mathcal{L}_{\|\nabla \mathcal{L}_r^\perp}$ and $\nabla_t \mathcal{L}_{\|\nabla \mathcal{L}_b^\perp}$, defined in Eq. (4.1). Then, $\mathbf{G}_t \subseteq \mathbf{K}_t^*$.*

Consequently, the DCGD algorithm defines the updated gradient denoted by $g_t^{\text{dual}}$ within $\mathbf{G}_t$ at each iteration $t$. A general framework for DCGD is presented in Algo 1.

---

**Algorithm 1** Dual Cone Gradient Descent (base)

---

**Require:** learning rate $\lambda$, max epoch $T$, initial point $\theta_0$
**for** $t = 1$ **to** $T$ **do**
    Choose $g_t^{\text{dual}} \in \mathbf{G}_t^*$
    $\theta_t = \theta_{t-1} - \lambda g_t^{\text{dual}}$
**end for**

---

### 4.2 Convergence Analysis

To discuss the convergence properties of DCGD, we introduce the concept of Pareto optimality (adapted to the PINN setting), which is a key in multi-objective optimization [46, 40].

**Definition 4.4.** (Pareto optimal and stationary) A point $\theta \in \mathbb{R}^d$ is said to be *Pareto-optimal* if there does not exist $\theta' \in \mathbb{R}^d$ such that

$$\mathcal{L}_r(\theta') \leq \mathcal{L}_r(\theta) \quad \text{and} \quad \mathcal{L}_b(\theta') \leq \mathcal{L}_b(\theta).$$

In addition, a point $\theta \in \mathbb{R}^d$ is said to be *Pareto-stationary* if there exists $\alpha_1, \alpha_2$ such that

$$\alpha_1 \nabla \mathcal{L}_r(\theta) + \alpha_2 \nabla \mathcal{L}_b(\theta) = 0, \quad \alpha_1, \alpha_2 \geq 0, \quad \alpha_1 + \alpha_2 = 1.$$

Intuitively, a Pareto-stationary point implies there is no feasible descent direction that would decrease all loss functions simultaneously. For example, consider a point $\theta_t$ at which the cosine of the angle $\phi_t$ between $\nabla \mathcal{L}_r(\theta_t)$ and $\nabla \mathcal{L}_b(\theta_t)$ is $-1$, i.e., $\cos(\phi_t) = -1$. Such a point is Pareto-stationary.

The following theorem guarantees the convergence of the DCGD algorithm proposed in Algo 1 under some regularities in a non-convex setting. Assume $\mathcal{L}(\theta^*) := \inf_{\theta \in \mathbb{R}^d} \mathcal{L}(\theta) > -\infty$.

**Theorem 4.5.** *Assume that both loss functions, $\mathcal{L}_b(\cdot)$ and $\mathcal{L}_r(\cdot)$, are differentiable and the total gradient $\nabla \mathcal{L}(\cdot)$ is L-Lipschitz continuous with $L > 0$. If $g_t^{dual}$ satisfies the following two conditions:*

*(i) $2\langle \nabla\mathcal{L}(\theta_t), g_t^{dual}\rangle - \|g_t^{dual}\|^2 \geq 0$,*

*(ii) There exists $M > 0$ such that $\|g_t^{dual}\| \geq M\|\nabla\mathcal{L}(\theta_t)\|$,*

*then, for $\lambda \leq \frac{1}{2L}$, DCGD in Algo. 1 converges to a Pareto-stationary point, or converges as*

$$\frac{1}{T+1}\sum_{t=0}^{T}\|\nabla\mathcal{L}(\theta_t)\|^2 \leq \frac{2\left(\mathcal{L}(\theta_0) - \mathcal{L}(\theta^*)\right)}{\lambda M(T+1)}. \tag{4.2}$$

Theorem 4.5 states that DCGD converges to either a Pareto-stationary point, characterized by $\phi_t$ such that $\cos(\phi_t) = -1$, or a stationary point at a rate of $\mathcal{O}(1/\sqrt{T})$ in the nonconvex setting. Unlike single-objective (nonconvex) optimization where the goal is to pursue a stationary point, in multi-objective optimization, it is ideal to find a Pareto-stationary point that balances all loss functions. Thus, DCGD offers significant theoretical and empirical advantages over popular optimization algorithms like SGD and ADAM, which are only guaranteed to converge to a stationary point. The convergence of DCGD to a Pareto-stationary point is empirically verified in Section 4.4.

## 4.3 Dual Cone Gradient Descent: Projection, Average, and Center

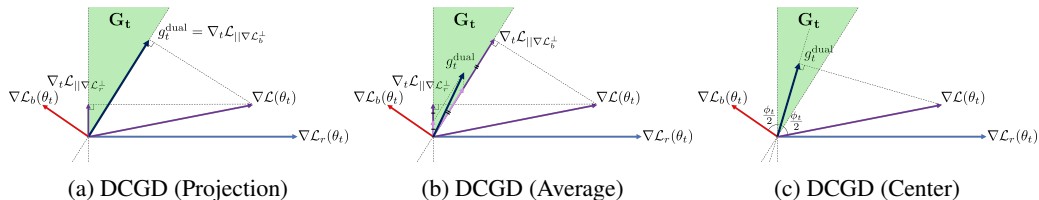

(a) DCGD (Projection)  (b) DCGD (Average)  (c) DCGD (Center)

Figure 4: The updated gradient $g_t^{\text{dual}}$ of three DCGD algorithms.

Different variants of DCGD can be designed by properly choosing the updated gradient $g_t^{\text{dual}}$ in $\mathbf{G}_t$ satisfying the conditions (i), (ii) of Theorem 4.5. We present three specific algorithms: projection, average, and center.

The first algorithm, named DCGD (Projection), uses the projection of the total gradient $\nabla\mathcal{L}(\theta_t)$ onto $\mathbf{G}_t$ when $\nabla\mathcal{L}(\theta_t) \notin \mathbf{K}_t^*$, which is the closest vector within $\mathbf{G}_t$ to $\nabla\mathcal{L}(\theta_t)$. Specifically, the DCGD (Projection) algorithm specifies $g_t^{\text{dual}}$ as follows: **(i)** $\nabla\mathcal{L}(\theta_t)$ if $\nabla\mathcal{L}(\theta_t) \in \mathbf{K}_t^*$, **(ii)** $\nabla_t\mathcal{L}_{\|\nabla\mathcal{L}_r^{\perp}}$ ($c_1 = 1, c_2 = 0$) if $\nabla\mathcal{L}(\theta_t) \notin \mathbf{K}_t^*$ and $\langle \nabla\mathcal{L}(\theta_t), \nabla\mathcal{L}_r(\theta_t)\rangle < 0$, **(iii)** $\nabla_t\mathcal{L}_{\|\nabla\mathcal{L}_b^{\perp}}$ ($c_1 = 0, c_2 = 1$) if $\nabla\mathcal{L}(\theta_t) \notin \mathbf{K}_t^*$ and $\langle \nabla\mathcal{L}(\theta_t), \nabla\mathcal{L}_b(\theta_t)\rangle < 0$. See also Eq. (E.1) and Algo. 2.

DCGD (Average) algorithm takes the average of $\nabla_t\mathcal{L}_{\|\nabla\mathcal{L}_r^{\perp}}$ and $\nabla_t\mathcal{L}_{\|\nabla\mathcal{L}_b^{\perp}}$ when the total gradient is outside $\mathbf{K}_t^*$, i.e., $c_1 = c_2 = \frac{1}{2}$ if $\nabla\mathcal{L}(\theta_t) \notin \mathbf{K}_t^*$. See Eq. (E.2) and Algo. 3.

We note that both DCGD (Projection) and DCGD (Average) use $\nabla\mathcal{L}(\theta_t)$ as $g_t^{\text{dual}}$ without any manipulation when $\nabla\mathcal{L}(\theta_t) \in \mathbf{K}_t^*$. Moreover, they require determining if the total gradient is contained in the dual cone at each iteration, which may incur additional computational costs. On the other hand, $g_t^{\text{dual}}$ of DCGD (Center) is given by

$$g_t^{\text{dual}} := \frac{\langle g_t^c, \nabla\mathcal{L}(\theta_t)\rangle}{\|g_t^c\|^2}g_t^c \text{ where } g_t^c = \frac{\nabla\mathcal{L}_b(\theta_t)}{\|\nabla\mathcal{L}_b(\theta_t)\|} + \frac{\nabla\mathcal{L}_r(\theta_t)}{\|\nabla\mathcal{L}_r(\theta_t)\|}, \tag{4.3}$$

which is geometrically interpreted as the projection of $\nabla\mathcal{L}(\theta_t)$ onto the angle bisector $g_t^c$ of $\nabla\mathcal{L}_r(\theta_t)$ and $\nabla\mathcal{L}_b(\theta_t)$. The following proposition shows that $g_t^{\text{dual}}$ of DCGD (Center) resides within $\mathbf{G}_t$

**Proposition 4.6.** *Consider the updated gradient $g_t^{dual}$ of DCGD (Center) defined in Eq. (4.3). Then, $g_t^{dual} \in \mathbf{G}_t$.*

The visualization of these three algorithms can be found in Figure 4 and their pseudocodes are provided in Appendix E. Moreover, the proposed DCGD algorithms satisfy the conditions (i) and (ii) of Theorem 4.5. Consequently, the following Corollary summarizes the convergence of the proposed DCGD algorithms.

**Corollary 4.7.** *We impose the same assumptions as in Theorem 4.5. Then, DCGD (Projection), DCGD (Average), and DCGD (Center) converge to either a Pareto-stationary point or a stationary point.*

In addition to the theoretical result in Corollary 4.7, Appendix D.1 provides an ablation study on the empirical performance of three specific algorithms for solving benchmark PDEs.

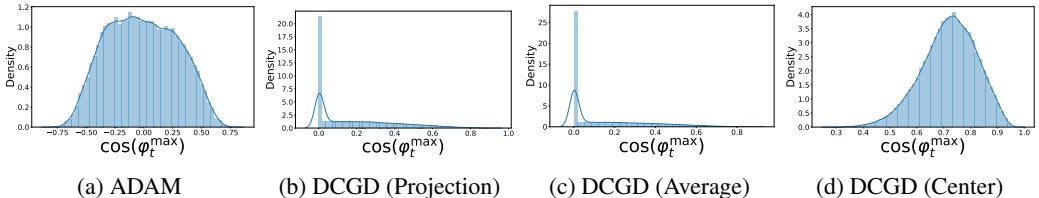

|     (a) ADAM     | (b) DCGD (Projection) | (c) DCGD (Average) | (d) DCGD (Center) |

Figure 5: Distribution of $\cos(\varphi_t^{\max})$ for each algorithm with $\varphi_t^{\max} = \max\{\varphi_t^r, \varphi_t^b\}$ where $\varphi_t^r$ is the angle between the updated vector and $\nabla\mathcal{L}_r(\theta_t)$, and $\varphi_t^b$ is the angle between the updated vector and $\nabla\mathcal{L}_b(\theta_t)$.

## 4.4 Benefits of the DCGD framework

This subsection discusses benefits of DCGD through illustrative examples. We first investigate how the proposed DCGD algorithms resolve the conflicting gradient issue discussed in Section 3. Given each algorithm, at each iteration $t$, we define $\varphi_t^r$ as the angle between the updated vector and $\nabla\mathcal{L}_r(\theta_t)$, and $\varphi_t^b$ as the angle between the updated vector and $\nabla\mathcal{L}_b(\theta_t)$. Also, let $\varphi_t^{\max} = \max\{\varphi_t^r, \varphi_t^b\}$. We highlight that both $\varphi_t^r$ and $\varphi_t^b$ are less than $\pi/2$ under DCGD algorithms, as they ensure that the updated vectors always belong to the dual cone. Figure 5 plots the distributions of $\cos(\varphi_t^{\max})$ for four different optimization algorithms: ADAM, DCGD (Projection), DCGD (Average), and DCGD (Center) during the training of PINNs for solving the Helmholtz equation. It shows that three DCGD algorithms completely eliminate conflicting gradients in contrast to ADAM. Moreover, we observe that the distributions of $\cos(\varphi_t^{\max})$ for DCGD (Projection) and DCGD (Average) are highly skewed toward zero, which implies that one of the two losses is unlikely to significantly improve. On the contrary, DCGD (Center) has a bell-shaped distribution with a mean of about 0.719, indicating that the two gradients are more aligned. This leads to a continuous reduction in both losses in a harmonious manner. Consistent with this observation, DCGD (Center) consistently outperforms DCGD (Projection) and DCGD (Average) in our experiments. Please refer to the ablation study D.1 for further comparisons.

We empirically demonstrate that DCGD can converge to a Pareto-stationary point. Consider a (slightly modified) toy example shown in [37, 41], which has two objective functions; see Appendix C.2 for more details. We solve the problem with 1,600 uniformly sampled initial points using ADAM, DCGD (Projection), DCGD (Average), and DCGD (Center). Then, we mark with a red dot the point at which the algorithm fails to reach a Pareto-stationary point. Figure 6 shows that while ADAM does not reach a Pareto-stationary point across many areas, all DCGD algorithms achieve convergence to Pareto-stationary points throughout the entire space.

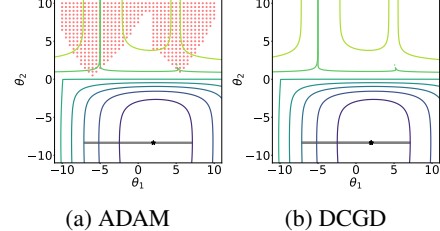

|  (a) ADAM  |  (b) DCGD  |

Figure 6: Toy example: the region where the algorithm fails to reach a Pareto-stationary point in multi-objective optimization

Several MTL algorithms, such as PCGrad [37], MGDA [40], CAGrad [41], Aligned-MTL [44], and Nash-MTL [43] have been developed based on different and independent approaches. In contrast, the proposed DCGD framework provides a principled solution to the problem of conflicting gradients by directly characterizing the dual cone. As a result, our framework unifies many of these MTL

algorithms as special cases, offering significant contributions not only to PINNs but also to the MTL domain. Proofs for the unification of MTL algorithms within the DCGD framework can be found in Appendix B.

## 5 Numerical Experiment

This section demonstrates the superiority of DCGD through three distinct perspectives. In Section 5.1, we compare the performance of DCGD on five benchmark equations with that of a range of methods, including ADAM [45], Learning Rate Annealing (LRA) [30], Neural Tangent Kernel (NTK) [31], PCGrad [37], MGDA [40], CAGrad [41], Aligned-MTL [44], MultiAdam [34], and DPM [35]. Section 5.2 shows that DCGD can provide more accurate solutions for failure modes of PINNs and complex PDEs where vanilla PINNs fail. In Section 5.3, we explore the compatibility of DCGD with existing loss balancing schemes such as LRA and NTK.

To compare the effectiveness of DCGD with other optimization algorithms, we measure the accuracy of the PINN solution trained by each optimizer using the relative $L^2$-error. Then, we run each experiment across 10 independent trials and report the mean, standard deviation, max, and min of the best accuracy.

### 5.1 Comparison on benchmark equations

We solve three popular benchmark equations (the Helmholtz equation, the viscous Burgers' equation, and the Klein-Gordon equation) and two high-dimensional PDEs (5D-Heat equation and 3D-Helmholtz equation) using vanilla PINNs with different optimization techniques. For DCGD, we employ an adaptive gradient version of the DCGD (Center) algorithm, the DCGD (Center) combined with ADAM (see Algo 5) by default for all experiments, provided in Appendix D.1. For other methods, we perform careful hyperparameter tuning based on the recommendations in their papers. The PDE equations and detailed experimental setting are provided in Appendix C.4. However, we do not report the performance of DPM because it is not only highly sensitive to hyperparameters but also exhibit poor performance, consistently observed in [47].

Table 1: Average of relative $L^2$ errors in 10 independent trials for each algorithm on three benchmark PDEs (3 independent trials for two high-dimensional PDEs). The value within the parenthesis indicates the standard deviation. '-' denotes that the optimizer failed to converge.

| | PDE equation | | | | |
|---|---|---|---|---|---|
| Optimizer | Helmholtz | Burgers' | Klein-Gordon | Heat (5D) | Helmholtz (3D) |
| ADAM | 0.0609 (0.0231) | 0.0683 (0.0285) | 0.0792 (0.0386) | 0.0097 (0.0072) | 0.6109 (0.2096) |
| LRA | 0.0066 (0.0025) | 0.0180 (0.0094) | 0.0069 (0.0037) | 0.0052 (0.0056) | 0.0831 (0.0123) |
| NTK | 0.0358 (0.0107) | 0.0224 (0.0061) | 0.0223 (0.0151) | 0.0027 (0.0012) | 0.4037 (0.2620) |
| PCGrad | 0.0109 (0.0031) | 0.0159 (0.0061) | 0.0286 (0.0064) | 0.0083 (0.0049) | 0.2532 (0.0476) |
| MGDA | 0.7590 (0.1180) | 0.9780 (0.0462) | 0.6690 (0.2790) | - | 0.9883 (0.0217) |
| CAGrad | 0.0735 (0.0390) | 0.0321 (0.0063) | 0.1850 (0.0301) | 0.0043 (0.0016) | 0.5854 (0.3032) |
| Aligned-MTL | 0.6570 (0.0805) | 0.0294 (0.0129) | 0.5571 (0.1824) | 0.0013 (0.0004) | 0.9138 (0.0645) |
| MultiAdam | 0.0211 (0.0032) | 0.0875 (0.0303) | 0.0228 (0.0038) | 0.0009 (0.0007) | 0.7809 (0.0031) |
| DCGD | **0.0029 (0.0005)** | **0.0124 (0.0046)** | **0.0069 (0.0027)** | **0.0008 (0.0003)** | **0.0774 (0.0250)** |
| DCGD+LRA | 0.0023 (0.0007) | 0.0104 (0.0021) | 0.0050 (0.0013) | 0.0012 (0.0005) | 0.1045 (0.0485) |
| DCGD+NTK | 0.0057 (0.0035) | 0.0113 (0.0040) | 0.0055 (0.0014) | 0.0009 (0.0004) | 0.3525 (0.2659) |

Table 1 displays the mean and standard deviation of the relative $L^2$ errors for each optimization algorithm applied to the three PDE equations. The error plots of approximated PINN solutions and other statistics of relative $L^2$ errors are summarized in Appendix C.4. In the result tables, we highlight **the best** and the second-best methods. While the second best methods vary across experiments, the proposed method consistently outperforms other algorithms achieving the lowest $L^2$ errors. This result underscores the robustness and adaptability of our method for solving various PDEs.

### 5.2 Failure mode of PINNs and Complex P(I)DEs

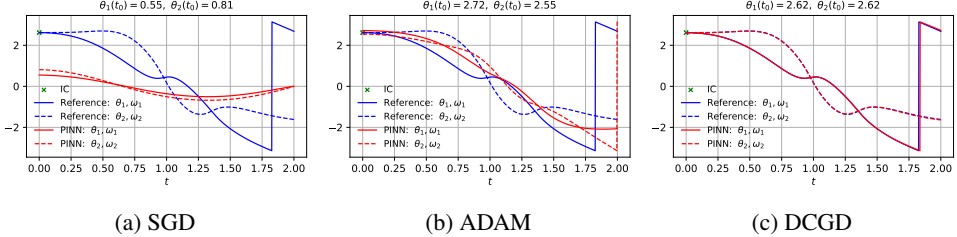

(a) SGD      (b) ADAM      (c) DCGD

Figure 7: Double pendulum problem: prediction of each method. SGD and ADAM find shifted solutions, but DCGD successfully approximates the reference solution.

We explore more challenging problems, including failure modes of PINNs and complex PDEs, where vanilla PINNs fail to approximate solutions, and highlight the universal applicability of DCGD. We refer to Appendix C.5 for detailed experimental settings.

First, we revisit the problem of a double pendulum in Steger et al. [32], which is highly sensitive to initial conditions. The goal is to solve the trajectory of

Table 2: Relative $L^2$ errors for DCGD (Center) on Chaotic KS equation, Convection equation and Volterra IDEs.

| Equation | Baseline | DCGD |
|---|---|---|
| Chaotic KS | 0.0687 | **0.0376** |
| Convection | 0.4880 | **0.0246** |
| Volterra IDEs | 0.0068 | **0.0011** |

$\{(\theta_1(t), \theta_2(t))\}_{t \geq t_0}$, governed by the nonlinear differential equation as discussed in Eq. (C.1). The reference solution and its first-order derivative are represented by the blue solid and dotted lines, respectively, in Figure 7. We train PINNs with SGD and ADAM to solve the double pendulum problem, where their solutions are depicted by the red solid and dotted lines in Figure 7a and Figure 7b, respectively. The PINN solutions trained with SGD and ADAM fail to accurately approximate the reference solution. In contrast, the reference solution is successfully recovered by our DCGD algorithm (see Figure 7c). Second, we present the performance of DCGD for two challenging PDEs: the chaotic Kuramoto-Sivashinsky (KS) equation and the convection equation. For the chaotic KS equation, we combine DCGD with the causal training scheme of [22], the current state-of-the art result. For the convection equation, DCGD is applied to PINNsFormer of [15]. As shown in Table 2, DCGD achieves the lowest relative $L^2$ errors for the complex PDEs compared to the existing optimally tuned strategies, demonstrating its effectiveness in overcoming failure modes of PINNs. Third, the universal applicability of DCGD is not limited to specific architectures, sampling techniques, and training schemes. For example, A-PINN, designed for solving integral equations and integro-differential equations, achieves state-of-the art results in nonlinear Volterra IDEs [9]. DCGD significantly improves the performance of A-PINN for solving Volterra IDEs, as shown in Table 2. Moreover, Table 4 shows that the performance of SPINN can be highly improved by applying DCGD for solving multi-dimensional PDEs.

### 5.3 Compatibility of DCGD with existing methods

The proposed DCGD framework can be easily combined with existing PINN training strategies, including loss balancing methods. To illustrate this advantage, we have designed DCGD algorithms that integrate with LRA and NTK, named DCGD (Center) + LRA and DCGD (Center) + NTK, respectively. Please refer to Algo. 6 for the detailed implementation.

We apply DCGD (Center) + LRA and DCGD (Center) + NTK to the same experiments described in Section 5.1. Tables 1 and 3 demonstrate that the performance of DCGD algorithms can be further enhanced across all the experiments in terms of the mean, maximum, and minimum of relative $L^2$ errors by integrating existing ideas from the literature.

## 6 Conclusion and Discussion

In this work, we provided a clear criterion for when PINNs might be adversely trained, in terms of the angle and relative magnitude ratio of the gradients of the PDE residual and boundary loss, through a geometric analysis. Based on this theoretical insight, we characterized a dual cone region where both losses can decrease simultaneously without gradient pathologies. We then proposed a general

framework for DCGD, which ensures that the updated gradient falls within the dual cone region, and provided a convergence analysis. Within this general framework, we introduced three specific DCGD algorithms and conduct extensive empirical experiments. Our experimental results demonstrate that the proposed DCGD algorithms outperform other optimization algorithms. In particular, DCGD is efficient in solving challenging problems such as failure modes of PINNs and complex PDEs compared to the current state-of-the art approaches. Furthermore, DCGD can be easily combined with other strategies and applied to variants of PINNs.

Although we have presented a novel optimization algorithm, DCGD, to address challenging issues in PINNs, there still remain some interesting and important questions. For instance, one could design a more powerful DCGD specification within the dual cone region that goes beyond the projection, average, and center techniques. Also, while we mainly consider multi-objective optimization for PINNs, future work can focus on more general and complex types of multi-task learning problems.

**Acknowledgement**

This work was supported by the National Research Foundation of Korea (NRF) grant funded by the Korea government (MSIT) (No.RS-2023-00253002), the Institute of Information & communications Technology Planning & Evaluation (IITP) grant funded by the Korea government (MSIT) (No.2020-0-01336, Artificial Intelligence Graduate School Program (UNIST)), and Startup Research Fund (1.220132.01) of UNIST (Ulsan National Institute of Science & Technology).

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

# A Proofs for Section 4

*Proof of Theorem 4.2.* Recall that $\phi_t$ is the angle between $\nabla\mathcal{L}_r(\theta_t)$ and $\nabla\mathcal{L}_b(\theta_t)$, and $R = \frac{\|\nabla\mathcal{L}_r(\theta_t)\|}{\|\nabla\mathcal{L}_b(\theta_t)\|}$ at each iteration $t$. Consider a cone $K_t$, defined as

$$\mathbf{K}_t := \{cx | c \geq 0, x \in \{\nabla\mathcal{L}_r(\theta_t), \nabla\mathcal{L}_b(\theta_t)\}\}.$$

**Case (i).** Suppose that $\langle\nabla\mathcal{L}_b(\theta_t), \nabla\mathcal{L}_r(\theta_t)\rangle \geq 0$. Observe that

$$\langle\nabla\mathcal{L}(\theta_t), \nabla\mathcal{L}_r(\theta_t)\rangle = \langle\nabla\mathcal{L}_b(\theta_t), \nabla\mathcal{L}_r(\theta_t)\rangle + \|\nabla\mathcal{L}_r(\theta_t)\|^2 \geq 0,$$
$$\langle\nabla\mathcal{L}(\theta_t), \nabla\mathcal{L}_b(\theta_t)\rangle = \langle\nabla\mathcal{L}_b(\theta_t), \nabla\mathcal{L}_r(\theta_t)\rangle + \|\nabla\mathcal{L}_b(\theta_t)\|^2 \geq 0.$$

Therefore, by the definition of the dual cone, we have $\nabla\mathcal{L}(\theta_t) \in \mathbf{K}_t^*$.

**Case (ii).** Suppose that $\langle\nabla\mathcal{L}_b(\theta_t), \nabla\mathcal{L}_r(\theta_t)\rangle < 0$ and $-\cos(\phi_t) \leq R \leq -\frac{1}{\cos(\phi_t)}$. By multiplying $\|\nabla\mathcal{L}_b\|\|\nabla\mathcal{L}_r\|$ to $-\cos(\phi_t) \leq R$, we get

$$-\cos(\phi_t) \leq R \Leftrightarrow -\|\nabla\mathcal{L}_b(\theta_t)\|\|\nabla\mathcal{L}_r(\theta_t)\|\cos(\phi_t) \leq \|\nabla\mathcal{L}_r(\theta_t)\|^2,$$
$$\Leftrightarrow \langle\nabla\mathcal{L}_b(\theta_t), \nabla\mathcal{L}_r(\theta_t)\rangle + \langle\nabla\mathcal{L}_r(\theta_t), \nabla\mathcal{L}_r(\theta_t)\rangle \geq 0,$$
$$\Leftrightarrow \langle\nabla\mathcal{L}(\theta_t), \nabla\mathcal{L}_r(\theta_t)\rangle \geq 0. \tag{A.1}$$

On the other hand, by multiplying $\|\nabla\mathcal{L}_b(\theta_t)\|^2\cos(\phi_t)$ to $R \leq -\frac{1}{\cos(\phi_t)}$, we have

$$R \leq -\frac{1}{-\cos(\phi_t)} \Leftrightarrow \|\nabla\mathcal{L}_b(\theta_t)\|\|\nabla\mathcal{L}_r(\theta_t)\|\cos(\phi_t) \geq -\|\nabla\mathcal{L}_b(\theta_t)\|^2,$$
$$\Leftrightarrow \langle\nabla\mathcal{L}_b(\theta_t), \nabla\mathcal{L}_r(\theta_t)\rangle + \langle\nabla\mathcal{L}_b(\theta_t), \nabla\mathcal{L}_b(\theta_t)\rangle \geq 0,$$
$$\Leftrightarrow \langle\nabla\mathcal{L}(\theta_t), \nabla\mathcal{L}_b(\theta_t)\rangle \geq 0. \tag{A.2}$$

Therefore, we conclude that if $\langle\nabla\mathcal{L}_b(\theta_t), \nabla\mathcal{L}_r(\theta_t)\rangle < 0$, then $-\cos(\phi_t) \leq R \leq -\frac{1}{\cos(\phi_t)}$ is equivalent to $\nabla\mathcal{L}(\theta_t) \in \mathbf{K}_t^*$.

□

*Proof of Proposition 4.3.* Recall that

$$\mathbf{G}_t := \left\{c_1\nabla_t\mathcal{L}_{\|\nabla\mathcal{L}_r^\perp} + c_2\nabla_t\mathcal{L}_{\|\nabla\mathcal{L}_b^\perp} | c_1, c_2 \geq 0\right\}. \tag{A.3}$$

It is enough to show that we have $\langle g, \nabla\mathcal{L}_r(\theta_t)\rangle \geq 0$ and $\langle g, \nabla\mathcal{L}_b(\theta_t)\rangle \geq 0$ for any $g \in \mathbf{G}_t$. By the definition of $\mathbf{G}_t$, there exists $c_1, c_2 \geq 0$ such that $g = c_1\nabla_t\mathcal{L}_{\|\nabla\mathcal{L}_r^\perp} + c_2\nabla_t\mathcal{L}_{\|\nabla\mathcal{L}_b^\perp}$ for all $g \in \mathbf{G}_t$. One can easily check that

$$\langle g, \nabla\mathcal{L}_r(\theta_t)\rangle = \langle c_2\nabla_t\mathcal{L}_{\|\nabla\mathcal{L}_b^\perp}, \nabla\mathcal{L}_r(\theta_t)\rangle$$
$$= \langle c_2\left(\nabla\mathcal{L}_r(\theta_t) - \frac{\langle\nabla\mathcal{L}_b(\theta_t), \nabla\mathcal{L}_r(\theta_t)\rangle}{\|\nabla\mathcal{L}_b(\theta_t)\|^2}\nabla\mathcal{L}_b(\theta_t)\right), \nabla\mathcal{L}_r(\theta_t)\rangle$$
$$= c_2\left(\|\nabla\mathcal{L}_r(\theta_t)\|^2 - \frac{|\langle\nabla\mathcal{L}_b(\theta_t), \nabla\mathcal{L}_r(\theta_t)\rangle|^2}{\|\nabla\mathcal{L}_b(\theta_t)\|^2}\right)$$
$$= c_2\|\nabla\mathcal{L}_r(\theta_t)\|^2(1 - \cos(\phi_t))$$
$$\geq 0$$

where $\phi_t$ is the angle between $\nabla\mathcal{L}_r(\theta_t)$ and $\nabla\mathcal{L}_b(\theta_t)$. One can derive that $\langle g, \nabla\mathcal{L}_b(\theta_t)\rangle \geq 0$ in the same manner. Therefore, we conclude that $\mathbf{G}_t \subset \mathbf{K}_t^*$.

□

*Proof of Theorem 4.5.* Let $\phi_t$ be the angle between $\nabla\mathcal{L}_r(\theta_t)$ and $\nabla\mathcal{L}_b(\theta_t)$, and $\psi_t$ be the angle between $g_t^{\text{dual}}$ and $\nabla\mathcal{L}(\theta_t)$ at the $t$-th iteration. Note that $\theta_{t+1} = \theta_t - \lambda g_t^{\text{dual}}$ where $g_t^{\text{dual}}$ satisfies the conditions (i), (ii) of Theorem 4.5.

First of all, DCGD algorithm reaches a Pareto-stationary point if $\phi_t = -1$ by Definition 4.4, at which the optimization process is stopped.

Otherwise, we first observe from the differentiability and $L$-Lipschitz continuity condition of $\nabla\mathcal{L}(\cdot)$ for all $x, y \in \mathbb{R}^d$:

$$
\begin{aligned}
\mathcal{L}(x) - \mathcal{L}(y) &= \int_0^1 \langle \nabla\mathcal{L}(y + t(x - y)),\ x - y \rangle \mathrm{d}t \\
&\leq \langle \nabla\mathcal{L}(y),\ x - y \rangle + \int_0^1 \langle \nabla\mathcal{L}(y + t(x - y)) - \nabla\mathcal{L}(y),\ x - y \rangle \mathrm{d}t \\
&\leq \langle \nabla\mathcal{L}(y),\ x - y \rangle + \int_0^1 \|\nabla\mathcal{L}(y + t(x - y)) - \nabla\mathcal{L}(y)\| \|x - y\| \mathrm{d}t \\
&\leq \langle \nabla\mathcal{L}(y),\ x - y \rangle + \int_0^1 Lt\|x - y\|^2 \mathrm{d}t \\
&= \langle \nabla\mathcal{L}(y),\ x - y \rangle + \frac{L}{2}\|x - y\|^2,
\end{aligned}
\tag{A.4}
$$

where we have used Cauchy-Schwarz inequality for the third inequality. Using Eq. (A.4) and Conditions (i), (ii) of Theorem 4.5, one calculates that for $\lambda \leq \frac{1}{2L}$,

$$
\begin{aligned}
\mathcal{L}(\theta_{t+1}) - \mathcal{L}(\theta_t) &\leq -\lambda \langle \nabla\mathcal{L}(\theta_t),\ g_t^{\mathrm{dual}} \rangle + \frac{L\lambda^2}{2}\|g_t^{\mathrm{dual}}\|^2 \\
&\leq -\lambda \langle \nabla\mathcal{L}(\theta_t),\ g_t^{\mathrm{dual}} \rangle + \frac{\lambda}{4}\|g_t^{\mathrm{dual}}\|^2 \\
&= -\frac{\lambda}{4} \left( 2\langle \nabla\mathcal{L}(\theta_t),\ g_t^{\mathrm{dual}} \rangle - \|g_t^{\mathrm{dual}}\|^2 + 2\langle \nabla\mathcal{L}(\theta_t),\ g_t^{\mathrm{dual}} \rangle \right) \\
&\leq -\frac{\lambda}{2} \langle \nabla\mathcal{L}(\theta_t),\ g_t^{\mathrm{dual}} \rangle \quad \because \text{ condition (i)} \\
&\leq -\frac{\lambda M}{2}\|\nabla\mathcal{L}(\theta_t)\|^2 \quad \because \text{ Cauchy-Swartz inequality and condition (ii)}
\end{aligned}
\tag{A.5}
$$

By using telescoping sums, we further obtain

$$
\sum_{t=0}^{T} \mathcal{L}(\theta_{t+1}) - \mathcal{L}(\theta_t) = \mathcal{L}(\theta_{T+1}) - \mathcal{L}(\theta_0)
$$

$$
\leq -\frac{\lambda M}{2} \sum_{t=0}^{T} \|\nabla\mathcal{L}(\theta_t)\|^2,
$$

which yields

$$
\frac{1}{T+1} \sum_{t=0}^{T} \|\nabla\mathcal{L}(\theta_t)\|^2 \leq \frac{2\left(\mathcal{L}(\theta_0) - \mathcal{L}(\theta_{T+1})\right)}{\lambda M(T+1)}
$$

$$
\leq \frac{2\left(\mathcal{L}(\theta_0) - \mathcal{L}(\theta^*)\right)}{\lambda M(T+1)}.
$$

$\square$

*Proof of Proposition 4.6.* Note that $g_t^c$ is the angle bisector of $\nabla\mathcal{L}_r(\theta_t)$ and $\nabla\mathcal{L}_b(\theta_t)$. From the formula of vector projection, $g_t^{\mathrm{dual}}$ of DCGD (Center) is the projection of $\nabla\mathcal{L}(\theta_t)$ on to $g_t^c$. Thus, it is enough to show that $g_t^c$ is included in $\mathbf{G}_t$.

We observe that

$$
\nabla_t \mathcal{L}_{\|\nabla\mathcal{L}_r^{\perp}} = \nabla\mathcal{L}(\theta_t) - \langle \nabla\mathcal{L}(\theta_t),\ \nabla\mathcal{L}_r(\theta_t) \rangle \frac{\nabla\mathcal{L}_r(\theta_t)}{\|\nabla\mathcal{L}_r(\theta_t)\|^2}
\tag{A.6}
$$

$$
= \nabla\mathcal{L}_b(\theta_t) - \langle \nabla\mathcal{L}_b(\theta_t),\ \nabla\mathcal{L}_r(\theta_t) \rangle \frac{\nabla\mathcal{L}_r(\theta_t)}{\|\nabla\mathcal{L}_r(\theta_t)\|^2},
\tag{A.7}
$$

and

$$\nabla_t \mathcal{L}_{\|\nabla \mathcal{L}_b^\perp} = \nabla \mathcal{L}(\theta_t) - \langle \nabla \mathcal{L}(\theta_t), \nabla \mathcal{L}_b(\theta_t) \rangle \frac{\nabla \mathcal{L}_b(\theta_t)}{\|\nabla \mathcal{L}_b(\theta_t)\|^2} \tag{A.8}$$

$$= \nabla \mathcal{L}_r(\theta_t) - \langle \nabla \mathcal{L}_r(\theta_t), \nabla \mathcal{L}_b(\theta_t) \rangle \frac{\nabla \mathcal{L}_b(\theta_t)}{\|\nabla \mathcal{L}_b(\theta_t)\|^2}. \tag{A.9}$$

Then, by defining $c_1 = \frac{1}{\|\nabla \mathcal{L}_b(\theta_t)\|(1-\cos(\phi_t))}$ and $c_2 = \frac{1}{\|\nabla \mathcal{L}_r(\theta_t)\|(1-\cos(\phi_t))}$, one can easily see that

$$c_1 \nabla_t \mathcal{L}_{\|\nabla \mathcal{L}_r^\perp} + c_2 \nabla_t \mathcal{L}_{\|\nabla \mathcal{L}_b^\perp} = \nabla \mathcal{L}_b(\theta_t) \left( c_1 - c_2 \frac{\langle \nabla \mathcal{L}_r(\theta_t), \nabla \mathcal{L}_b(\theta_t) \rangle}{\|\nabla \mathcal{L}_b(\theta_t)\|^2} \right)$$

$$+ \nabla \mathcal{L}_r(\theta_t) \left( c_2 - c_1 \frac{\langle \nabla \mathcal{L}_r(\theta_t), \nabla \mathcal{L}_b(\theta_t) \rangle}{\|\nabla \mathcal{L}_r(\theta_t)\|^2} \right)$$

$$= \frac{\nabla \mathcal{L}_b(\theta_t)}{\|\nabla \mathcal{L}_b(\theta_t)\|} + \frac{\nabla \mathcal{L}_r(\theta_t)}{\|\nabla \mathcal{L}_r(\theta_t)\|}$$

$$= g_t^c.$$

That is, $g_t^c$ can be expressed as $c_1 \nabla_t \mathcal{L}_{\|\nabla \mathcal{L}_r^\perp} + c_2 \nabla_t \mathcal{L}_{\|\nabla \mathcal{L}_b^\perp}$ for some $c_1, c_2 \geq 0$. Therefore, $g_t^c$ is in $\mathbf{G}_t$. $\qquad \square$

*Proof of Corollary 4.7.* We will show that $g_t^{\text{dual}}$ of each DCGD algorithm satisfies the conditions (i), (ii) of Theorem 4.5. Three algorithms are summarized in Algo. 2, Algo. 3, and Algo. 4. We note that a conflict threshold $\alpha$ is introduced as a stopping condition for DCGD algorithms, as they can reach a Pareto-stationary point characterized by $\phi_t = \pi$. That is, the algorithm stops when the parameter converges close to a Pareto-stationary point such that $\pi - \alpha < \phi_t \leq \pi$. Here, we assume $\alpha \geq 0$ is fixed.

**1. DCGD (Projection):** Note that it is trival to show that $g_t^{\text{dual}} = \nabla \mathcal{L}(\theta_t)$, when $\langle \nabla \mathcal{L}(\theta_t), \nabla \mathcal{L}_b(\theta_t) \rangle \geq 0$, satisfies the conditions (i), (ii). Thus, we focus on the case when $\langle \nabla \mathcal{L}(\theta_t), \nabla \mathcal{L}_b(\theta_t) \rangle < 0$.

First of all, we need to show the condition (i)

$$2 \langle \nabla \mathcal{L}(\theta_t), g_t^{\text{dual}} \rangle - \|g_t^{\text{dual}}\|^2 = \|\nabla \mathcal{L}(\theta_t)\|^2 - \|g_t^{\text{dual}} - \nabla \mathcal{L}(\theta_t)\|^2 \geq 0,$$

which is equivalent to that $\|\nabla \mathcal{L}(\theta_t)\| \geq \|g_t^{\text{dual}} - \nabla \mathcal{L}(\theta_t)\|$. Using Eq. (A.6), one directly calculates that

$$\|2 \nabla_t \mathcal{L}_{\|\mathcal{L}_r^\perp} - \nabla \mathcal{L}(\theta_t)\|^2 = \left\| \nabla \mathcal{L}(\theta_t) - \langle \nabla \mathcal{L}(\theta_t), \nabla \mathcal{L}_r(\theta_t) \rangle \frac{\nabla \mathcal{L}_r(\theta_t)}{\|\nabla \mathcal{L}_r(\theta_t)\|^2} \right\|^2$$

$$= \|\nabla \mathcal{L}(\theta_t)\|^2.$$

In the same manner, we have $\|2 \nabla_t \mathcal{L}_{\|\mathcal{L}_b^\perp} - \nabla \mathcal{L}(\theta_t)\|^2 = \|\nabla \mathcal{L}(\theta_t)\|^2$. Since $g_t^{\text{dual}}$ is chosen in $\mathbf{G}_t$, specifically $c_1 = 1, c_2 = 0$ or $c_1 = 0, c_2 = 0$, we can write

$$\|g_t^{\text{dual}} - \nabla \mathcal{L}(\theta_t)\| = \left\| \left( c_1 \nabla \mathcal{L}_{\|\mathcal{L}_r^\perp}(\theta_t) + c_2 \nabla \mathcal{L}_{\|\mathcal{L}_b^\perp}(\theta_t) \right) - \nabla \mathcal{L}(\theta_t) \right\|$$

$$\leq \left\| \left( c_1 \nabla \mathcal{L}_{\|\mathcal{L}_r^\perp}(\theta_t) + c_2 \nabla \mathcal{L}_{\|\mathcal{L}_b^\perp}(\theta_t) \right) - \frac{c_1 + c_2}{2} \nabla \mathcal{L}(\theta_t) \right\| + \left\| \left( \frac{c_1 + c_2}{2} - 1 \right) \nabla \mathcal{L}(\theta_t) \right\|$$

$$\leq \left\| \frac{c_1}{2} \left( 2 \nabla \mathcal{L}_{\|\mathcal{L}_r^\perp}(\theta_t) - \nabla \mathcal{L}(\theta_t) \right) \right\| + \left\| \frac{c_2}{2} \left( 2 \nabla \mathcal{L}_{\|\mathcal{L}_b^\perp}(\theta_t) - \nabla \mathcal{L}(\theta_t) \right) \right\| + \left\| \left( \frac{c_1 + c_2}{2} - 1 \right) \nabla \mathcal{L}(\theta_t) \right\|$$

$$= \frac{c_1}{2} \|\nabla \mathcal{L}(\theta_t)\| + \frac{c_2}{2} \|\nabla \mathcal{L}(\theta_t)\| + \left| \frac{c_1 + c_2}{2} - 1 \right| \|\nabla \mathcal{L}(\theta_t)\|$$

$$= \left( \frac{c_1 + c_2}{2} + \left| \frac{c_1 + c_2}{2} - 1 \right| \right) \|\nabla \mathcal{L}(\theta_t)\| \tag{A.10}$$

$$= \|\nabla \mathcal{L}(\theta_t)\|.$$

where we have used $c_1 + c_2 = 1$ for obtaining the last inequality. Therefore, the condition (i) is satisfied.

We further suppose that $\langle \nabla \mathcal{L}(\theta_t), \nabla \mathcal{L}_b(\theta_t) \rangle < 0$. Then, $g_t^{\text{dual}} = \nabla_t \mathcal{L}_{\| \mathcal{L}_b^\perp}$. Let $\phi_t$ be the angle between $\nabla \mathcal{L}_b(\theta_t)$ and $\nabla \mathcal{L}_r(\theta_t)$, and $\psi_t$ be the angle between $g_t^{\text{dual}}$ and $\nabla \mathcal{L}(\theta_t)$. Note that $\phi_t \leq \pi - \alpha$ where $\alpha$ is conflict threshold. Otherwise, the algorithm stops when $\pi - \alpha < \phi_t \leq \pi$ (see Algo. 2). Then, since $\psi_t = \phi_t - \frac{\pi}{2}$, we have

$$
\begin{aligned}
\| g_t^{\text{dual}} \| &= \| \nabla_t \mathcal{L}_{\| \mathcal{L}_b^\perp} \| \\
&= \| \nabla \mathcal{L}(\theta_t) \| \cos(\psi_t) \\
&= \| \nabla \mathcal{L}(\theta_t) \| \cos \left( \phi_t - \frac{\pi}{2} \right) \\
&\geq \| \nabla \mathcal{L}(\theta_t) \| \cos \left( \frac{\pi}{2} - \alpha \right).
\end{aligned}
$$

Thus, by choosing $M = \cos \left( \frac{\pi}{2} - \alpha \right)$, the condition (ii) is satisfied. We repeat the same analysis for the case when $\langle \nabla \mathcal{L}(\theta_t), \nabla \mathcal{L}_b(\theta_t) \rangle < 0$.

**2. DCGD (Average):** Similarly to DCGD (Projection), we focus on the case where DCGD (Average) specifies $c_1 = c_2 = \frac{1}{2}$, given by

$$
g_t^{\text{dual}} = \frac{1}{2} \left( \nabla_t \mathcal{L}_{\| \nabla \mathcal{L}_r^\perp} + \nabla_t \mathcal{L}_{\| \nabla \mathcal{L}_b^\perp} \right),
$$

when $\nabla \mathcal{L}(\theta_t) \notin \mathbf{K}_t^*$. Eq A.10 with $c_1 = c_2 = 1/2$ directly leads to

$$
\| g_t^{\text{dual}} - \nabla \mathcal{L}(\theta_t) \| \leq \| \nabla \mathcal{L}(\theta_t) \|,
$$

implying that the condition (i) is satisfied.

Next, suppose $\langle \nabla \mathcal{L}(\theta_t), \nabla \mathcal{L}_b(\theta_t) \rangle < 0$. Then, the condition (ii) is satisfied with $M = \frac{1}{2} \cos \left( \frac{\pi}{2} - \alpha \right)$ since

$$
\begin{aligned}
\| g_t^{\text{dual}} \| &= \frac{1}{2} \left\| \nabla_t \mathcal{L}_{\| \mathcal{L}_r^\perp} + \nabla_t \mathcal{L}_{\| \mathcal{L}_b^\perp} \right\| \\
&\geq \frac{1}{2} \| \nabla_t \mathcal{L}_{\| \mathcal{L}_b^\perp} \| \\
&= \frac{1}{2} \| \nabla \mathcal{L}(\theta_t) \| \cos \left( \phi_t - \frac{\pi}{2} \right) \\
&\geq \frac{1}{2} \cos \left( \frac{\pi}{2} - \alpha \right) \| \nabla \mathcal{L}(\theta_t) \|.
\end{aligned}
$$

When $\langle \nabla \mathcal{L}(\theta_t), \nabla \mathcal{L}_r(\theta_t) \rangle < 0$, the condition (ii) is also satisfied with $M = \frac{1}{2} \cos \left( \frac{\pi}{2} - \alpha \right)$.

**3. DCGD (Center):** the updated vector of DCGD (Center) is given by

$$
g_t^{\text{dual}} = \frac{\langle g_t^c, \nabla \mathcal{L}(\theta_t) \rangle}{\| g_t^c \|^2} g_t^c
$$

where $g_t^c = \frac{\nabla \mathcal{L}_b(\theta_t)}{\| \nabla \mathcal{L}_b(\theta_t) \|} + \frac{\nabla \mathcal{L}_r(\theta_t)}{\| \nabla \mathcal{L}_r(\theta_t) \|}$. Since $g_t^{\text{dual}}$ is the angle bisector of $\nabla \mathcal{L}_r(\theta_t)$ and $\nabla \mathcal{L}_b(\theta_t)$ (see Proof of Proposition 4.6), $\psi_t$, the angle between $\nabla \mathcal{L}(\theta_t)$ and $g_t^{\text{dual}}$, is less or equal to $\phi_t / 2$, i.e., $\psi_t \leq \frac{\phi_t}{2} \leq \frac{\pi}{2}$. From the fact that $g_t^{\text{dual}}$ is the projection of $\nabla \mathcal{L}(\theta_t)$ onto $g_t^c$, we have

$$
\begin{aligned}
\| \nabla \mathcal{L}(\theta_t) - g_t^{\text{dual}} \| &= \| \nabla \mathcal{L}(\theta_t) \| \sin(\psi_t) \\
&\leq \| \nabla \mathcal{L}(\theta_t) \| \sin \left( \frac{\phi_t}{2} \right) \\
&\leq \| \nabla \mathcal{L}(\theta_t) \| \sin \left( \frac{\pi - \alpha}{2} \right) \\
&\leq \| \nabla \mathcal{L}(\theta_t) \|,
\end{aligned}
$$

and

$$\|g_t^{\text{dual}}\| = \|\nabla \mathcal{L}(\theta_t)\| \cos(\psi_t)$$

$$\geq \|\nabla \mathcal{L}(\theta_t)\| \cos\left(\frac{\phi_t}{2}\right)$$

$$\geq \|\nabla \mathcal{L}(\theta_t)\| \cos\left(\frac{\pi - \alpha}{2}\right).$$

Consequently, the conditions (i) and (ii) are satisfied for DCGD (Center).

$\square$

*Remark* A.1. Suppose that one employs a decaying scheme for the conflict threshold $\alpha_t$ such that $\alpha_t = \mathcal{O}(t^{-\gamma})$ with where $0 \leq \gamma < 1$, for example, $\alpha_t = t^{-\gamma}$. In this case, the convergence rate of the DCGD algorithm to a stationary point becomes $\mathcal{O}\left(\frac{1}{T^{1-\gamma}}\right)$, as $M$ in condition (ii) may depend on the conflict threshold $\alpha_t$. For all our experiments, we set $\alpha$ to be fixed.

# B  Unification of MTL algorithms within the DCGD framework

In this section, we prove that several MTL algorithms can be understood as special cases of the DCGD framework under the PINN's formulation.

*Proof.* **1. MGAD [40]:** The updated gradient $g_t^{\text{MGDA}}$ of MGDA is defined by selecting the minimum-norm element from the convex combinations of $\nabla \mathcal{L}_r(\theta_t)$ and $\nabla \mathcal{L}_b(\theta_t)$ if there is gradient conflict $\langle \nabla \mathcal{L}_r(\theta_t), \nabla \mathcal{L}_b(\theta_t) \rangle < 0$:

$$g_t^{\text{MGDA}} := \underset{\alpha_1, \alpha_2 \geq 0}{\arg\min} \|u\|, \qquad \text{s.t. } u = \alpha_1 \nabla \mathcal{L}_r(\theta_t) + \alpha_2 \nabla \mathcal{L}_b(\theta_t), \alpha_1 + \alpha_2 = 1.$$

One can easily show that $\langle g_t^{\text{MGDA}}, g_t^{\text{MGDA}} \rangle = \langle g_t^{\text{MGDA}}, \nabla \mathcal{L}_r(\theta_t) \rangle = \langle g_t^{\text{MGDA}}, \nabla \mathcal{L}_b(\theta_t) \rangle \geq 0$. Thus,

$$g_t^{\text{MGDA}} \in \mathbf{K}_t^*.$$

**2. PCGrad [37]:** PCGrad uses the same update direction with DCGD (Average) when $\langle \nabla \mathcal{L}_r(\theta_t), \nabla \mathcal{L}_b(\theta_t) \rangle < 0$,

$$g_t^{\text{PCGrad}} = \frac{1}{2}\left(\nabla_t \mathcal{L}_{\|\nabla \mathcal{L}_r^\perp} + \nabla_t \mathcal{L}_{\|\nabla \mathcal{L}_b^\perp}\right) \in \mathbf{K}_t^*.$$

and takes $\nabla \mathcal{L}_r(\theta_t) + \nabla \mathcal{L}_b(\theta_t)$ when $\langle \nabla \mathcal{L}_r(\theta_t), \nabla \mathcal{L}_b(\theta_t) \rangle \geq 0$. The latter case is also contained in $\mathbf{K}_t^*$. Therefore, $g_t^{\text{PCGrad}} \in \mathbf{K}_t^*$.

**3. Nash-MTL [43]:** Nash-MTL considers a Nash bargaining solution to balance the loss gradients. The update gradient $g_t^{\text{Nash-MTL}}$ is be defined by

$$g_t^{\text{Nash-MTL}} := G_t v_t, \tag{B.1}$$

$$\text{s.t. } G_t^\top G_t v_t = v_t^{-1}. \tag{B.2}$$

where $G_t = [\nabla \mathcal{L}_r(\theta_t), \nabla \mathcal{L}_b(\theta_t)]$. We can find the solution $v_t$ satisfying Eq. (B.2) as following. By letting $v_t = \begin{bmatrix} v_1 \\ v_2 \end{bmatrix}$, we have

$$\begin{bmatrix} \|\nabla \mathcal{L}_r(\theta_t)\|^2 & \langle \nabla \mathcal{L}_r(\theta_t), \nabla \mathcal{L}_b(\theta_t) \rangle \\ \langle \nabla \mathcal{L}_r(\theta_t), \nabla \mathcal{L}_b(\theta_t) \rangle & \|\nabla \mathcal{L}_b(\theta_t)\|^2 \end{bmatrix} \begin{bmatrix} v_1 \\ v_2 \end{bmatrix} = \begin{bmatrix} \frac{1}{v_1} \\ \frac{1}{v_2} \end{bmatrix},$$

which is equivalent to

$$\begin{cases} \|\nabla \mathcal{L}_r(\theta_t)\|^2 v_1^2 + \langle \nabla \mathcal{L}_r(\theta_t), \nabla \mathcal{L}_b(\theta_t) \rangle v_1 v_2 = 1 \\ \|\nabla \mathcal{L}_b(\theta_t)\|^2 v_2^2 + \langle \nabla \mathcal{L}_r(\theta_t), \nabla \mathcal{L}_b(\theta_t) \rangle v_1 v_2 = 1 \end{cases}. \tag{B.3}$$

Therefore, we can derive $v_2 = \frac{\|\nabla\mathcal{L}_r(\theta_t)\|}{\|\nabla\mathcal{L}_b(\theta_t)\|}v_1$. Substituting $v_2 = \frac{\|\nabla\mathcal{L}_r(\theta_t)\|}{\|\nabla\mathcal{L}_b(\theta_t)\|}v_1$ back into the first equation of Eq. (B.3) leads to

$$\|\nabla\mathcal{L}_r(\theta_t)\|^2 v_1^2 + \|\nabla\mathcal{L}_r(\theta_t)\|^2 \cos(\phi_t) v_1^2 = 1$$

$$\Leftrightarrow v_1 = \sqrt{\frac{1}{1+\cos\phi_t}}\frac{1}{\|\nabla\mathcal{L}_r(\theta_t)\|}$$

$$\Leftrightarrow v_2 = \frac{\|\nabla\mathcal{L}_r(\theta_t)\|}{\|\nabla\mathcal{L}_b(\theta_t)\|}v_1 = \sqrt{\frac{1}{1+\cos\phi_t}}\frac{1}{\|\nabla\mathcal{L}_b(\theta_t)\|}$$

$$\Leftrightarrow G_t v_t = \sqrt{\frac{1}{1+\cos\phi_t}}\left(\frac{\nabla\mathcal{L}_r(\theta_t)}{\|\nabla\mathcal{L}_r(\theta_t)\|} + \frac{\nabla\mathcal{L}_b(\theta_t)}{\|\nabla\mathcal{L}_b(\theta_t)\|}\right)$$

where $\phi_t$ is the angle between $\nabla\mathcal{L}_r(\theta_t)$ and $\nabla\mathcal{L}_b(\theta_t)$. Thus, the update gradient $g_t^{\text{Nash-MTL}}$ has same direction with DCGD (Center). That is, $g_t^{\text{Nash-MTL}} \in \mathbf{K}_t^*$.

$\square$

## C  Experimental details

### C.1  Software and hardware environments

We conduct all experiments with PYTHON 3.10.9 and PYTORCH 1.13.1, CUDA 11.6.2, NVIDIA Driver 510.10 on Ubuntu 22.04.1 LTS server which equipped with AMD Ryzen Threadripper PRO 5975WX, NVIDIA A100 80GB and NVIDIA RTX A6000.

### C.2  Toy example

We slightly modify the toy example in [41] to show our proposed method can expand the region of initial points that converge to the Pareto set. Consider the following loss functions with $\theta = (\theta_1, \theta_2) \in \mathbb{R}^2$:

$$L_0(\theta) = L_1(\theta) + L_2(\theta) \text{ where}$$
$$L_1(\theta) = 2c_1(\theta)f_1(\theta) + c_2(\theta)g_1(\theta) \text{ and } L_2(\theta) = c_1(\theta)f_2(\theta) + c_2(\theta)g_2(\theta),$$
$$f_1(\theta) = \log(\max(0.5(-\theta_1 - 7) - \tanh(-\theta_2), 0.000005)) + 6,$$
$$f_2(\theta) = \log(\max(0.5(-\theta_1 + 3) - \tanh(-\theta_2) + 2, 0.000005)) + 6,$$
$$g_1(\theta) = ((-\theta_1 + 7)^2 + 0.1 \cdot (-\theta_2 - 8)^2)/10 - 20,$$
$$g_2(\theta) = ((-\theta_1 - 7)^2 + 0.1 \cdot (-\theta_2 - 8)^2)/10 - 20,$$
$$c_1(\theta) = \max(\tanh(0.5 \cdot \theta_2), 0),$$
$$c_2(\theta) = \max(\tanh(-0.5 \cdot \theta_2), 0).$$

The landscape and contour map of above loss function are shown in Figure 8. The Pareto set is highlighted in gray in Figure 8b. We solve the above problem using ADAM, DCGD (Projection), DCGD (Average), DCGD (Center) for 100,000 epochs with different initial points. The initial points are selected as $1,600$ uniform grid points within $[-10, 10] \times [-10, 10]$. Then, we mark with a red dot the point at which the optimizer fails to converge to the Pareto set.

### C.3  Details for Figure 2, Figure 1, and Figure 5

In this experiment, we use the 7-layer fully connected neural network with 20 neurons per layer and a hyperbolic tangent activation function. We train PINN models using SGD with the learning rate of $0.01$ for $10,000$ epochs. In addition, 100 data points are sampled in boundaries and 10,000 points in the domain.

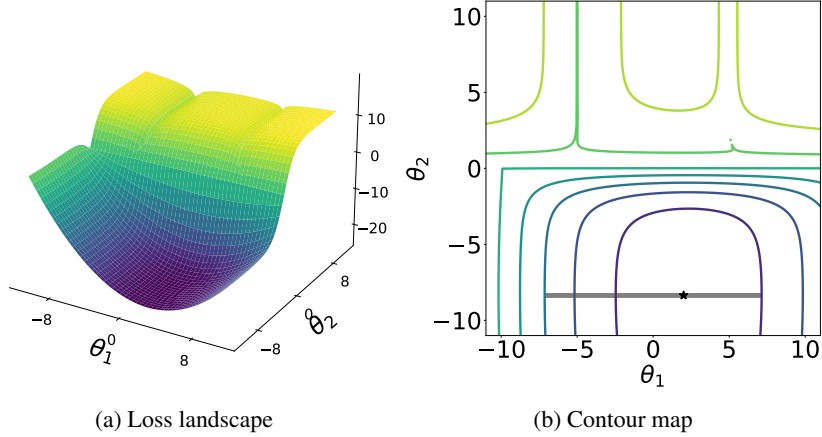

(a) Loss landscape          (b) Contour map

Figure 8: The loss landscape and contour map of the toy example.

## C.4 Details for Section 5.1

**Benchmark equations**   We consider Helmholtz equation, viscous Burgers' equation, and Klein-Gordon equation as the benchmark equations.

The Helmholtz equation is described by

$$\Delta u(x,y) + k^2 u(x,y) = f(x,y), \quad (x,y) \in \Omega,$$
$$u(x,y) = 0, \quad (x,y) \in \partial\Omega,$$
$$\Omega = [-1,1] \times [-1,1].$$

The solution is given by $u^*(x,y) = \sin(a_1\pi x)\sin(a_2\pi y)$ where

$$f(x,y) = (k^2 - a_1^2\pi^2 - a_2^2\pi^2)\sin(a_1\pi x)\sin(a_2\pi y)$$

In our experiment, we choose parameters: $k = 1, a_1 = 1, a_2 = 4$ as in [30].

The Viscous Burgers' equation is given by

$$u_t(t,x) + uu_x(t,x) - \nu u_{xx}(t,x) = 0, (x,t) \in [0,1] \times \Omega,$$
$$u(0,x) = -\sin(\pi x), x \in \Omega,$$
$$u(t,-1) = u(t,1) = 0, t \in [0,1],$$
$$\Omega = [-1,1]$$

where $\nu = \frac{0.01}{\pi}$.

The Klein-Gordon equation is

$$\Delta u(t,x) + \gamma u^k(t,x) = f(t,x), (t,x) \in [0,T] \times \Omega,$$
$$u(0,x) = g_1(x), x \in \Omega$$
$$u_t(0,x) = g_2(x), x \in \Omega$$
$$u(t,x) = h(t,x), (t,x) \in [0,T] \times \partial\Omega$$
$$\Omega = [0,1]$$

We set parameters to $k = 3, \gamma = 1, T = 1$ and the initial conditions, $g_1(x) = x, g_2(x) = 0$ for all $x \in \Omega$ following [30]. Then we can use the solution $u^*(t,x) = x\cos(5\pi t) + (tx)^3$ where $f(t,x)$ is derived by given equation .

We employ a 3-layer fully connected neural network with 50 neurons per layer and use the hyperbolic tangent activation function for all experiments in Section 5.1. At each iteration, 128 points are randomly sampled in boundaries and 10 times more points in the domain as the collocation points. We just randomly sample the points in the boundaries if there exists an analytic solution, otherwise the points were resampled from a pre-generated set for each iteration. More specifically, for the case

of Viscous Burger's equation, there is pre-determined 456 boundary points and we randomly sample in this set of points. We train PINNs for 50,000 epochs with Glorot normal initialization [48] using DCGD algorithms, ADAM [45], LRA [30], NTK [31], PCGrad [37], MultiAdam [34], and DPM [35].

We search for the initial learning rate among $\lambda = \{10^{-3}, 10^{-4}, 10^{-5}\}$ and use a exponential decay scheduler with a decay rate of $0.9$ and a decay step $= 1,000$. For ADAM, we use the default parameters: $\beta_1 = 0.9, \beta_2 = 0.999, \epsilon = 10^{-8}$ as in [45]. For LRA, we set $\alpha = 0.1$, which is the best hyperparameter reported in [30]. For MultiAdam, we use $\beta_1, \beta_2 = 0.99$ as recommended in [34]. For DPM, we test $\delta = \{10^{-1}, 10^{-2}, 10^{-3}\}, \epsilon = \{10^{-1}, 10^{-2}, 10^{-3}\}, w = \{1, 1.01, 1.001\}$.

To compute the effectiveness of various optimization algorithms, we evaluate the accuracy of the PINN solutions $u(\cdot; \theta)$ using the relative $L^2$-error defined as:

$$\text{Relative } L^2 \text{ error } = \frac{\sqrt{\sum_{i=1}^{N} |u(\boldsymbol{x}_i; \theta) - u(\boldsymbol{x}_i)|^2}}{\sqrt{\sum_{i=1}^{N} |u(\boldsymbol{x}_i)|^2}}$$

where $u(\cdot)$ is the true solution and $\{\boldsymbol{x}_i\}_{i=1}^{N}$ is the set of test samples. Unless the equation has an analytic solution, we use the numerical reference solution for $u(\boldsymbol{x})$, which solved by finite element method [1].

In Table 3, we report the best and worst-case relative $L^2$ errors of each method across 10 independent trials (3 independent trials for two high-dimensional PDEs).

| Equation | Helmholtz | | Burgers' | | Klein-Gordon | | Heat (5D) | | Helmholtz (5D) | |
|---|---|---|---|---|---|---|---|---|---|---|
| Optimizer | Max | Min | Max | Min | Max | Min | Max | Min | Max | Min |
| ADAM | 0.1053 | 0.0315 | 0.1413 | 0.0413 | 0.1586 | 0.0376 | 0.01985 | 0.0046 | 0.8990 | 0.4072 |
| LRA | 0.0108 | 0.0032 | 0.0391 | **0.0080** | 0.0166 | **0.0037** | 0.0131 | 0.0011 | **0.0991** | 0.0693 |
| NTK | 0.0532 | 0.0225 | 0.0358 | 0.0148 | 0.0581 | 0.0078 | 0.0044 | 0.0016 | 0.7743 | 0.2177 |
| PCGrad | 0.0170 | 0.0070 | 0.0322 | 0.0091 | 0.0399 | 0.0156 | 0.0149 | 0.0034 | 0.2907 | 0.1861 |
| MGDA | 1.0000 | 0.3441 | 1.0617 | 0.9037 | 1.0245 | 0.2168 | - | - | 1.0053 | 0.9577 |
| CAGrad | 0.1550 | 0.0330 | 0.0485 | 0.0235 | 0.2845 | 0.0872 | 0.0063 | 0.0024 | 1.0070 | 0.3070 |
| Aligned-MTL | 0.7784 | 0.5062 | 0.0640 | 0.0152 | 0.9133 | 0.2922 | 0.0018 | 0.0007 | 1.0001 | 0.8453 |
| MultiAdam | 0.0249 | 0.0149 | 0.1506 | 0.0537 | 0.0273 | 0.0160 | 0.0018 | **0.0003** | 0.7843 | 0.7768 |
| DCGD (Center) | **0.0038** | **0.0019** | **0.0016** | 0.0096 | **0.0112** | 0.0042 | **0.0012** | 0.0006 | 0.1007 | **0.0428** |
| DCGD (Center) + LRA | 0.0036 | 0.0013 | 0.0150 | 0.0056 | 0.0068 | 0.0036 | 0.0019 | 0.0007 | 0.1686 | 0.0515 |
| DCGD (Center) + NTK | 0.0157 | 0.0033 | 0.0175 | 0.0065 | 0.0079 | 0.0035 | 0.0015 | 0.0006 | 0.7276 | 0.1411 |

Table 3: Maximum and minimum of relative $L^2$ errors in 10 independent trials (3 independent trials for two high-dimensional PDEs) for each algorithm.

We plot the exact solution, PINN solution, and its error for each benchmark equation in Figures 10, 11, and 9

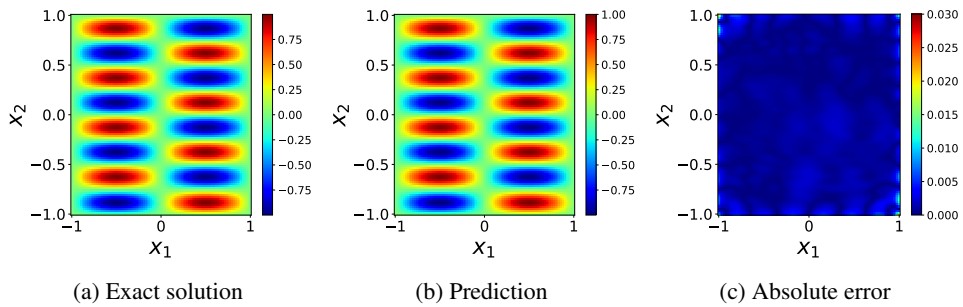

(a) Exact solution        (b) Prediction        (c) Absolute error

Figure 9: Helmholtz equation: approximated solution versus the reference solution.

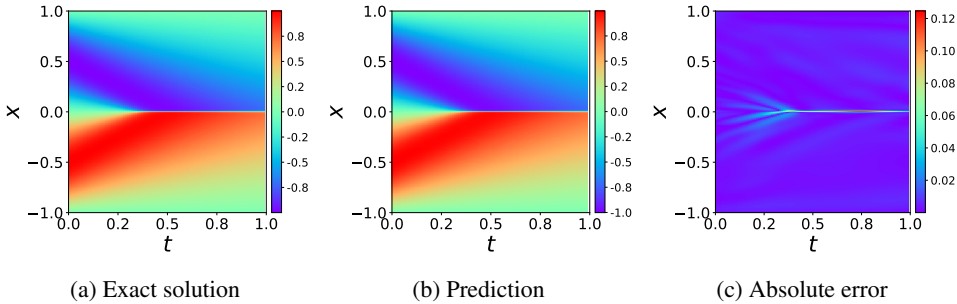

(a) Exact solution           (b) Prediction           (c) Absolute error

Figure 10: Burgers' equation: approximated solution versus the reference solution.

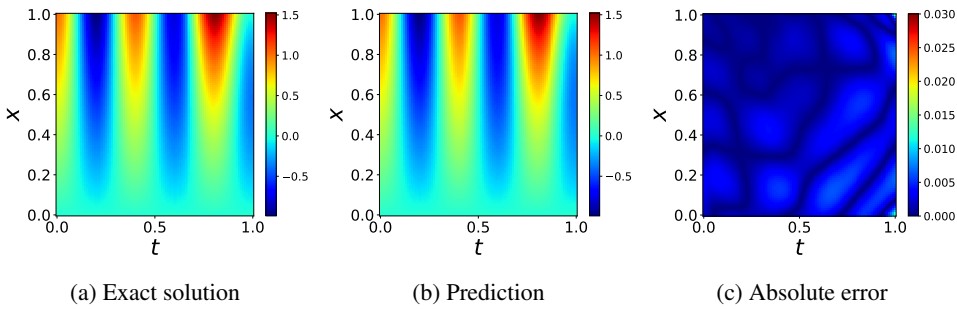

(a) Exact solution           (b) Prediction           (c) Absolute error

Figure 11: Klein-Gordon equation: approximated solution versus the reference solution.

**High-dimensional equations**   We consider the following 3-dimensional Helmholtz equation

$$\Delta u(x,y,z) + k^2 u(x,y,z) = f(x,y,z), \qquad\qquad (x,y,z) \in \Omega,$$
$$u(x,y,z) = 0, \qquad\qquad (x,y,z) \in \partial\Omega,$$
$$\Omega = [-1,1]^3.$$

The solution is given by $u^*(x,y) = \sin(a_1\pi x)\sin(a_2\pi y)\sin(a_3\pi z)$ where

$$f(x,y,z) = (k^2 - a_1^2\pi^2 - a_2^2\pi^2 - a_3^2\pi^2)\sin(a_1\pi x)\sin(a_2\pi y)\sin(a_3\pi z)$$

with $k = 1, a_1 = 4, a_2 = 4, a_3 = 3$.

We employ a 5-layer fully connected neural network with 128 neurons per layer and use the hyperbolic tangent activation function. At each iteration, 128 points are randomly sampled in boundaries and 500 times more points in the domain as the collocation points. We train PINNs for 30,000 epochs with Glorot normal initialization.

We use initial learning rate among $\lambda = 10^{-3}$ and use a exponential decay scheduler with a decay rate of 0.9 and a decay step $= 1,000$.

For 5-dimensional Heat equation, we follow the experiment setting in Hao et al. [49]. The PDE can be expressed as following:

$$u_t = k\Delta u + f(x,t), \qquad\qquad x \in \Omega \times [0,1]$$
$$\mathbf{n} \cdot \nabla u = g(x,t), \qquad\qquad x \in \partial\Omega \times [0,1]$$
$$u(x,0) = g(x,0), \qquad\qquad x \in \Omega$$

where the geometric domain $\Omega = \{x : ||x|| \leq 1\}$ and

$$f(x,t) := -\frac{1}{d}||x||^2 \exp\left(-\frac{1}{2}||x||^2 + t\right)$$
$$g(x,t) := \exp\left(-\frac{1}{2}||x||^2 + t\right)$$

## C.5  Details for Section 5.2

**Double pendulum problem**  Consider the double pendulum which have two point mass pendulums with masses $m_1, m_2$, two rod with length $l_1, l_2$. Let $\theta_1, \theta_2$ is the angle that the pendulums each make with the vertical and $\Delta\theta = \theta_1 - \theta_2$. Set the gravitational acceleration $g = 9.81$. (see Figure 12)

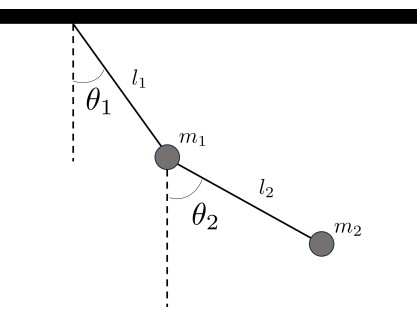

Figure 12: Simple double pendulum example

Then the dynamics of double pendulum can be described by following nonlinear differential equation system with $y = [\theta_1, \theta_2]^T$:

$$y'' = \begin{bmatrix} f_1(y, y') \\ f_2(y, y') \end{bmatrix} \text{ subject to } y(t_0) = \begin{bmatrix} \theta_1(t_0) \\ \theta_2(t_0) \end{bmatrix}, y'(t_0) = \begin{bmatrix} \omega_1(t_0) \\ \omega_2(t_0) \end{bmatrix}, \tag{C.1}$$

where

$$\omega_1 = \dot{\theta}_1,$$

$$\omega_2 = \dot{\theta}_2,$$

$$f_1(y, y'') = \frac{m_2 l_1 \omega_1^2 \sin(2\Delta\theta) + 2m_2 l_2 \omega_2^2 \sin\Delta\theta + 2gm_2 \cos\theta_2 \sin\Delta\theta + 2gm_1 \sin\theta_1}{2l_1(m_1 + m_2 \sin^2\Delta\theta)},$$

$$f_2(y, y'') = \frac{m_2 l_2 \omega_2^2 \sin(2\Delta\theta) + 2(m_1 + m_2)l_1 \omega_1^2 \sin\Delta\theta + 2g(m_1 + m_2)\cos\theta_1 \sin\Delta\theta}{2l_2(m_1 + m_2 \sin^2\Delta\theta)}.$$

In this experiment, the initial conditions are $\theta_1(t_0) = \theta_2(t_0) = \theta_0 = 150°$, $\omega_1(t_0) = \omega_2(t_0) = 0$.

We replicate the experiment setup done in [32]. We use a six-layer feed-forward network of 30 neurons on each layer with the swish activation function. We train this model with ADAM optimizer with the default hyperparameters for 20,000 epochs. We also train the same model using DCGD (Center). Figure 13 shows the training curves of ADAM and DCGD.

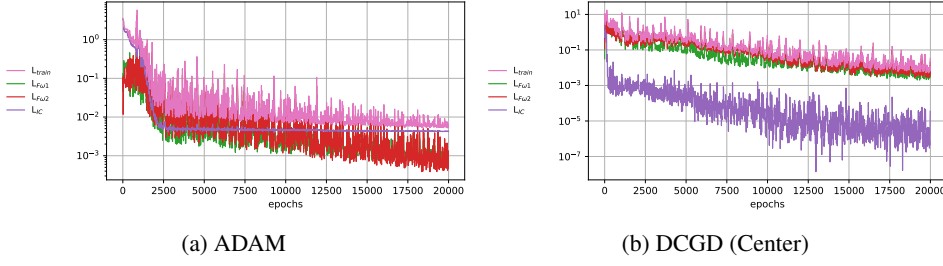

(a) ADAM          (b) DCGD (Center)

Figure 13: Loss trajectory of each method in the double pendulum problem.

**Convection equation**  We train PINNsFormer in Liu et al. [15] to solve convection equation which can expressed as following:

$$u_t + \beta u_x = 0, \quad \forall x \in [0, 2\pi], t \in [0, 1]$$
$$u(x, 0) = \sin(x),$$
$$u(0, t) = u(2\pi, t)$$

Where $\beta = 50$. we follow the default setting of [15] and train the model by 500 epochs.

**chotic Kuramoto-Sivashinsky equation**    We use causal training in Wang et al. [22] to solve chaotic Kuramoto-Sivashinsky equation. We use 5 layers modifed-MLP with 64 neurons per layer. and train this model 50,000 epochs for each tolerance.

$$u_t + \alpha u u_x + \beta u_{xx} + \gamma u_{xxxx} = 0, \quad \forall x \in [0, 2\pi], t \in [0, 0.5]$$
$$u(0, x) = \cos(x)(1 + \sin(x))$$

where $\alpha = 100/16, \beta = 100/16^2, \gamma = 100/16^4$.

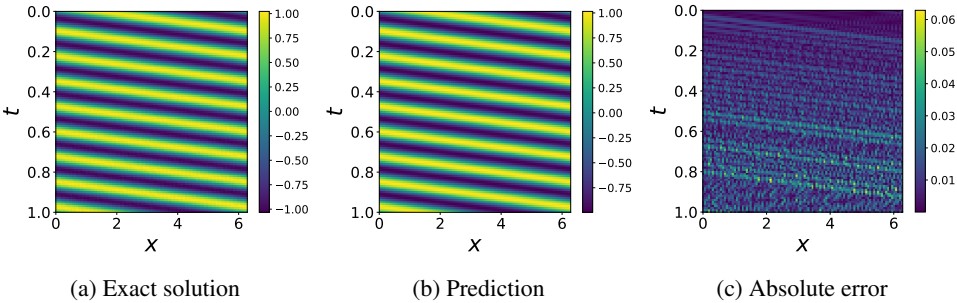

(a) Exact solution      (b) Prediction      (c) Absolute error

Figure 14: Convection equation: approximated solution versus the reference solution.

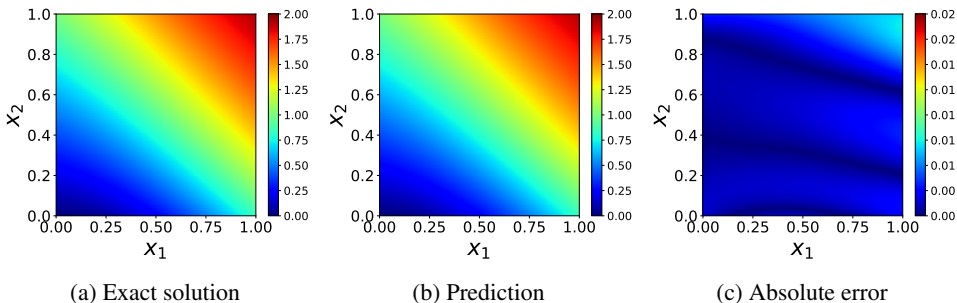

(a) Exact solution      (b) Prediction      (c) Absolute error

Figure 15: 2D-Volterra equation: approximated solution versus the reference solution.

**Auxiliary-PINN: Nonlinear integro-differential equation**    A-PINN is a variant of PINNs, designed to solve integro-differential equations [9]. We apply our DCGD algorithms to A-PINN for solving the following nonlinear 2-dimensional Volterra IDE:

$$\frac{\partial^2 u(t, x)}{\partial t^2} = \frac{\partial u(t, x)}{\partial x} - \frac{\partial u(t, x)}{\partial t} - u(t, x) + g(t, x) + \lambda \int_0^x \int_0^t f \cos(y_1 - y_2) u(y_1, y_2) dy_1 dy_2$$

where the boundary conditions are $u(0, x) = x, \frac{\partial u(0, x)}{\partial t} = \sin(x)$, and $u(t, 0) = t \sin(t), 0 \leq t, x \leq 1$. The analytic solution is $u^*(t, x) = x + t \sin(t + x)$ with $\lambda = 1$ where $g(t, x)$ is derived by given equation.

Within the framework of A-PINN, it converts the above integral equation into the following equation by representing integrals as auxiliary output variables:

$$\frac{\partial^2 u(t, x)}{\partial t^2} = \frac{\partial u(t, x)}{\partial x} - \frac{\partial u(t, x)}{\partial t} - u(t, x) + g(t, x) + \lambda v(t, x),$$
$$\frac{\partial v(t, x)}{\partial x} = \int_0^t f \cos(y_1 - x) u(y_1, x) dy_1 = t w(t, x),$$
$$\frac{\partial w(t, x)}{\partial t} = \cos(t - x) u(t, x),$$

where the new variables $v$ and $w$ satisfies the boundary condition $v(t, 0) = 0, w(0, x) = 0$.

For A-PINNs, we employ a 3-layer fully connected neural network with 50 neurons per layer and a hyperbolic tangent activation function. For training, 128 points are randomly sampled in boundaries and 10 times more points in the domain as the collocation points in each epochs. We train A-PINN models for $5,000$ epochs.

**Separable PINN: 3-dimensional Helmholtz equation**   SPINN is a a novel architecture designed to effectively reduce the computational cost of PINNs, especially when addressing high-dimensional PDEs [17]. To test the performance of DCGD for SPINNs, we consider the following 3-dimensional Helmholtz equation

$$\Delta u(x,y,z) + k^2 u(x,y,z) = f(x,y,z), \quad (x,y,z) \in \Omega,$$
$$u(x,y,z) = 0, \quad (x,y,z) \in \partial\Omega,$$
$$\Omega = [-1,1]^3.$$

The solution is given by $u^*(x,y) = \sin(a_1\pi x)\sin(a_2\pi y)\sin(a_3\pi z)$ where

$$f(x,y,z) = (k^2 - a_1^2\pi^2 - a_2^2\pi^2 - a_3^2\pi^2)\sin(a_1\pi x)\sin(a_2\pi y)\sin(a_3\pi z)$$

with $k = 1, a_1 = 4, a_2 = 4, a_3 = 3$.

We follow the optimal hyperparameter setting reported in [17]. For ADAM and DCGD (Center), the learning rate is $0.001$. The input points are resampled every 100 epochs. Regarding model architecture, we use the SOTA model, so called (SPINN + Modified MLP). We record the mean and standard deviation of relative $L^2$ errors from 3 independent trials in Table 4, indicating that the performance of SPINN can be significantly improved when trained with DCGD for a varying number of collocation points.

| Method | $N_c$ | Relative $L^2$ error | Training speed |
|---|---|---|---|
| SPINN | $16^3$ | 0.0578 (0.0039) | 1.65 (ms/iter) |
| | $32^3$ | 0.0352 (0.0035) | 1.78 (ms/iter) |
| | $64^3$ | 0.0280 (0.0066) | 2.38 (ms/iter) |
| | $128^3$ | 0.0294 (0.0123) | 2.71 (ms/iter) |
| | $256^3$ | 0.0319 (0.0026) | 5.12 (ms/iter) |
| SPINN + DCGD (Center) | $16^3$ | 0.0447 (0.0176) | 1.76 (ms/iter) |
| | $32^3$ | 0.0104 (0.0048) | 1.90 (ms/iter) |
| | $64^3$ | 0.0032 (0.0002) | 2.59 (ms/iter) |
| | $128^3$ | **0.0015 (0.0003)** | 2.85 (ms/iter) |
| | $256^3$ | 0.0020 (0.0009) | 5.34 (ms/iter) |

Table 4: Helmholtz Equation (3d): Relative $L^2$ errors and training speed. $N_c$ is the number of collocation points.

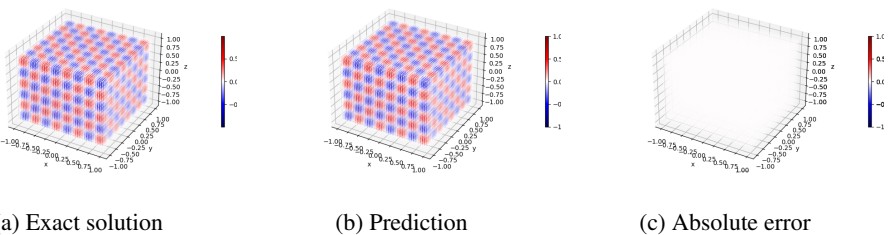

(a) Exact solution          (b) Prediction          (c) Absolute error

Figure 16: 3D-Helmholtz equation: approximated solution versus the reference solution.

# D Supplemental results

## D.1 Ablation study

In Section 4.3, we introduce three specific algorithms: DCGD (Projection), DCGD (Average), and DCGD (Center). We conduct an ablation study to investigate the impact of different updated gradient schemes within the dual cone region. More specifically, we compare the performance of these three algorithms on five benchmark equations. Detailed experimental settings can be found in Appendices C.4.

Tables 5 and 6 demonstrate that DCGD (Center) outperforms the other DCGD algorithms across all experiments. Therefore, we consider DCGD (Center) as the default DCGD algorithm.

| Equation | Helmholtz | Burgers' | Klein-Gordon | Heat (5D) | Helmholtz (3D) |
|---|---|---|---|---|---|
| Optimizer | Mean (std) | Mean (std) | Mean (std) | Mean (std) | Mean (std) |
| Projection | 0.0089 (0.0022) | 0.0139 (0.0035) | 0.0216 (0.0130) | 0.0074 (0.0049) | 0.1251 (0.0044) |
| Average | 0.0166 (0.0124) | 0.0156 (0.0032) | 0.0292 (0.0088) | 0.0051 (0.0008) | 0.3161 (0.0911) |
| Center | **0.0029 (0.0005)** | **0.0124 (0.0046)** | **0.0069 (0.0027)** | **0.0008 (0.0003)** | **0.0774 (0.0250)** |

Table 5: Average and standard deviation of relative $L^2$ errors in 10 independent trials (3 independent trials for two high-dimensional PDEs) for each DCGD algorithm.

| Equation | Helmholtz | | Burgers' | | Klein-Gordon | | Heat (5D) | | Helmholtz (3D) | |
|---|---|---|---|---|---|---|---|---|---|---|
| Optimizer | Max | Min | Max | Min | Max | Min | Max | Min | Max | Min |
| Projection | 0.0138 | 0.0062 | 0.0217 | 0.0097 | 0.0573 | 0.0120 | 0.0144 | 0.0036 | 0.1286 | 0.1189 |
| Average | 0.0469 | 0.0078 | 0.0209 | 0.0115 | 0.0442 | 0.0178 | 0.0060 | 0.0040 | 0.3942 | 0.1883 |
| Center | **0.0038** | **0.0019** | **0.0163** | **0.0096** | **0.0112** | **0.0042** | **0.0163** | **0.0096** | **0.1007** | **0.0428** |

Table 6: Min and Max Relative $L^2$ errors in 10 independent trials (3 independent trials for two high-dimensional PDEs) for each DCGD algorithm.

## D.2 Computational cost

In this section, we acknowledge that our proposed method incurs higher computational costs due to the need for backpropagation for each individual loss. Nonetheless, through a comparison of training speeds, we empirically demonstrate that DCGD achieves superior performance with computational costs comparable to those of existing competitors.

| | PDE equation | |
|---|---|---|
| Optimizer | Heat (5D) | Helmholtz (3D) |
| ADAM | 11.1 (iter/s) | 9.05 (iter/s) |
| L-BFGS | 0.53 (iter/s) | 1.19 (iter/s) |
| LRA | 3.39 (iter/s) | 5.43 (iter/s) |
| NTK | 3.92 (iter/s) | 7.49 (iter/s) |
| MultiAdam | 4.10 (iter/s) | 6.85 (iter/s) |
| PCGrad | 5.98 (iter/s) | 8.94 (iter/s) |
| MGDA | 3.46 (iter/s) | 6.16 (iter/s) |
| CAGrad | 4.22 (iter/s) | 8.80 (iter/s) |
| Aligned-MTL | 4.10 (iter/s) | 7.97 (iter/s) |
| DCGD (Average) | 3.78 (iter/s) | 5.42 (iter/s) |
| DCGD (Projection) | 3.70 (iter/s) | 5.64 (iter/s) |
| DCGD (Center) | 4.35 (iter/s) | 8.90 (iter/s) |

Table 7: Training speed in higher dimensional equations example

# E   Pseudo codes of algorithms

This section provides pseudo codes for the proposed DCGD algorithms.

Firstly, DCGD (Projection) uses the projection of the total gradient $\nabla\mathcal{L}(\theta_t)$ onto $\mathbf{G}_t$ when $\nabla\mathcal{L}(\theta_t) \notin \mathbf{K}_t^*$. Otherwise, $\nabla\mathcal{L}(\theta_t)$ is used. Then the update vector $g_t^{\text{dual}}$ can be defined as follow:

$$
\text{DCGD (Projection)} \qquad g_t^{\text{dual}} = \begin{cases} \nabla\mathcal{L}(\theta_t), & \text{if } \nabla\mathcal{L}(\theta_t) \in \mathbf{K}_t^* \\ \nabla_t\mathcal{L}_{\|\nabla\mathcal{L}_r^\perp}, & \text{if } \nabla\mathcal{L}(\theta_t) \notin \mathbf{K}_t^* \text{ and } \langle\nabla\mathcal{L}(\theta_t), \nabla\mathcal{L}_r(\theta_t) < 0\rangle \\ \nabla_t\mathcal{L}_{\|\nabla\mathcal{L}_b^\perp}. & \text{if } \nabla\mathcal{L}(\theta_t) \notin \mathbf{K}_t^* \text{ and } \langle\nabla\mathcal{L}(\theta_t), \nabla\mathcal{L}_b(\theta_t) < 0\rangle \end{cases}
$$
(E.1)

Secondly, DCGD (Average) uses the the average of $\nabla_t\mathcal{L}_{\|\nabla\mathcal{L}_r^\perp}$ and $\nabla_t\mathcal{L}_{\|\nabla\mathcal{L}_b^\perp}$ when the total gradient is outside $\mathbf{K}_t^*$. Otherwise, $\nabla\mathcal{L}(\theta_t)$ is used. The update vector $g_t^{\text{dual}}$ of DCGD (Average) is defined as follow:

$$
\text{DCGD (Average)} \qquad g_t^{\text{dual}} = \begin{cases} \nabla\mathcal{L}(\theta_t) & \text{if } \nabla\mathcal{L}(\theta_t) \in \mathbf{K}_t^* \\ \frac{1}{2}(\nabla_t\mathcal{L}_{\|\nabla\mathcal{L}_r^\perp} + \nabla_t\mathcal{L}_{\|\nabla\mathcal{L}_b^\perp}), & \text{if } \nabla\mathcal{L}(\theta_t) \notin \mathbf{K}_t^* \end{cases}
$$
(E.2)

Thirdly, DCGD (Center) employs the following update vector $g_t^{\text{dual}}$, regardless of whether the total gradient $\nabla\mathcal{L}(\theta_t)$ is included in $\mathbf{K}_t^*$:

$$
\text{DCGD (Center)} \qquad g_t^{\text{dual}} = \frac{\langle g_t^c, \nabla\mathcal{L}(\theta_t)\rangle}{\|g_t^c\|^2} g_t^c \text{ where } g_t^c = \frac{\nabla\mathcal{L}_b(\theta_t)}{\|\nabla\mathcal{L}_b(\theta_t)\|} + \frac{\nabla\mathcal{L}_r(\theta_t)}{\|\nabla\mathcal{L}_r(\theta_t)\|}
$$
(E.3)

Here, pseudo codes of these algorithms are summarized in Algorithms 2, 3, and 4. Note that we introduce a conflict threshold $\alpha$ as a stopping condition for DCGD algorithms, as they can reach a Pareto-stationary point characterized by $\phi_t = \pi$. That is, the algorithm stops when the parameter converges close to a Pareto-stationary point such that $|\cos(\phi_t) - \pi| < \alpha$. Throughout our experiments, we set $\alpha = 10^{-8}$.

---
**Algorithm 2** DCGD (Projection)

---

**Require:** learning rate $\lambda$, max epoch $T$, initial point $\theta_0$, gradient threshold $\varepsilon$, conflict threshold $\alpha$
**for** $t = 1$ **to** $T$ **do**
   **if** $\pi - \alpha < \phi_t \leq \pi$ or $\|\nabla\mathcal{L}(\theta_t)\| < \varepsilon$ **then**
      **break**
   **end if**
   **if** $\nabla\mathcal{L}(\theta_t) \notin \mathbf{K}^*$ **then**
      $g_t^{dual} = \nabla\mathcal{L}(\theta_t)$
   **else if** $\nabla\mathcal{L}(\theta_t) \in \mathbf{K}^*$ and $\langle\nabla\mathcal{L}(\theta_t), \nabla\mathcal{L}_r(\theta_t)\rangle < 0$ **then**
      $g_t^{dual} = \nabla_t\mathcal{L}_{\|\nabla\mathcal{L}_r^\perp}$
   **else if** $\nabla\mathcal{L}(\theta_t) \in \mathbf{K}^*$ and $\langle\nabla\mathcal{L}(\theta_t), \nabla\mathcal{L}_b(\theta_t)\rangle < 0$ **then**
      $g_t^{dual} = \nabla_t\mathcal{L}_{\|\nabla\mathcal{L}_b^\perp}$
   **end if**
**end for**

---

DCGD algorithms can be easily combined with other optimizers or strategies thanks to its flexible framework. For example, one can design a DCGD algorithm combined with ADAM to leverage advantages of adaptive gradient methods, see Algo. 5. For our experiments, we consider the DCGD (center) combined with ADAM as the default. Algorithm 6 presents the psedo code for DCGD combined with a loss balancing method such as LRA and NTK.

---

**Algorithm 3** DCGD (Average)

---

**Require:** learning rate $\lambda$, max epoch $T$, initial point $\theta_0$, gradient threshold $\varepsilon$, conflict threshold $\alpha$
  **for** $t = 1$ **to** $T$ **do**
    **if** $\pi - \alpha < \phi_t \leq \pi$ or $\|\nabla\mathcal{L}(\theta_t)\| < \varepsilon$ **then**
      **break**
    **end if**
    **if** $\nabla\mathcal{L}(\theta_t) \notin \mathbf{K}^*$ **then**
      $g_t^{dual} = \frac{1}{2}\nabla_t\mathcal{L}_{\|\nabla\mathcal{L}_r^\perp} + \frac{1}{2}\nabla_t\mathcal{L}_{\|\nabla\mathcal{L}_b^\perp}$
    **else**
      $g_t^{dual} = \nabla\mathcal{L}(\theta_t)$
    **end if**
    $\theta_t = \theta_{t-1} - \lambda g_t^{dual}$
  **end for**

---

 

---

**Algorithm 4** DCGD (Center)

---

**Require:** learning rate $\lambda$, max epoch $T$, initial point $\theta_0$, gradient threshold $\varepsilon$, conflict threshold $\alpha$
  **for** $t = 1$ **to** $T$ **do**
    **if** $\pi - \alpha < \phi_t \leq \pi$ or $\|\nabla\mathcal{L}(\theta_t)\| < \varepsilon$ **then**
      **break**
    **end if**
    $g_t^c = \frac{\nabla\mathcal{L}_b(\theta_t)}{\|\nabla\mathcal{L}_b(\theta_t)\|} + \frac{\nabla\mathcal{L}_r(\theta_t)}{\|\nabla\mathcal{L}_r(\theta_t)\|}$
    $g_t^{dual} = \frac{\langle g_t^c, \nabla\mathcal{L}(\theta_t)\rangle}{\|g_t^c\|^2} g_t^c$
    $\theta_t = \theta_{t-1} - \lambda g_t^{dual}$
  **end for**

---

 

---

**Algorithm 5** DCGD with ADAM

---

**Require:** learning rate $\lambda$, max epoch $T$, betas $\beta_1, \beta_2$, DCGD operator $\mathrm{DCGD}(\cdot)$
  **for** $t = 1$ **to** $T$ **do**
    $g_t^{dual} = \mathrm{DCGD}(\mathcal{L}_r(\theta), \mathcal{L}_b(\theta))$
    $g_t \leftarrow g_t^{dual}$
    $m_t \leftarrow \beta_1 m_{t-1} + (1 - \beta_1)g_t$
    $v_t \leftarrow \beta_2 v_{t-1} + (1 - \beta_2)g_t^2$
    $\widehat{m}_t \leftarrow \frac{m_t}{1 - \beta_1^t}$
    $\theta_t \leftarrow \theta_{t-1} - \gamma_t \frac{\widehat{m}_t}{\sqrt{\widehat{v}_t} + \epsilon}$
  **end for**

---

 

---

**Algorithm 6** DCGD with a loss balancing method

---

**Require:** learning rate $\lambda$, max epoch $T$, loss balancing operator $\mathrm{LB}(\cdot)$
  **for** $t = 1$ **to** $T$ **do**
    $(\beta_r, \beta_b)_t = \mathrm{LB}(\mathcal{L}_r(\theta_t), \mathcal{L}_b(\theta_t))$
    $\mathcal{L}_b(\theta_t) \leftarrow \beta_b \mathcal{L}_b(\theta_t)$
    $\mathcal{L}_r(\theta_t) \leftarrow \beta_r \mathcal{L}_r(\theta_t)$
    Choose $g_t^{dual} \in \mathbf{K}_t^*$
    $\theta_t = \theta_{t-1} - \lambda g_t^{dual}$
  **end for**

---

