# OpenReview forum: "Dual Cone Gradient Descent for Training Physics-Informed Neural Networks"
_NeurIPS.cc/2024/Conference — NeurIPS 2024 poster_

### Official Review · Reviewer_SdDT · 2024-07-03

**Soundness:** 3
**Presentation:** 3
**Contribution:** 3
**Rating:** 7
**Confidence:** 4

**Summary:**

**Summary:** The paper proposes viewing the optimization of PINNs as a multi-objective optimization problem, with boundary and residual terms as potentially conflicting objective functions. To address conflicting gradients in the optimization process, the dual cone of the cone generated by the gradients of the boundary and residual terms is designated as the set of non-conflicting update directions. Several algorithms that guarantee updates from this dual cone are proposed, and their empirical performance is assessed.

**General Impression:** The manuscript is well-written, easy to follow, and generally of high quality. The numerical experiments are convincing and demonstrate the good performance of the proposed method. However, I disagree to a certain extent with the multi-objective viewpoint for the optimization of PINNs. For well-posed forward problems, the boundary loss and the interior loss are not really conflicting losses—at least if we assume that the network ansatz is sufficiently expressive. I expand more on this issue below in the questions. Nevertheless, the preprint presents a convincing suite of numerical experiments that seem to benefit from the multi-objective optimization approach.

**Strengths:**

**Strengths:**

1. Considering the dual cone of the interior and residual gradients is a simple and sound idea. The presentation is clear and understandable.
2. The section on numerical results demonstrates the benefits of the proposed method and incorporates many recently proposed techniques for the optimization of PINNs.

**Weaknesses:**

See questions.

**Questions:**

**Questions**

1. It is unclear whether PINN optimization should be viewed as a multi-objective optimization problem. Please comment.
2. I am missing a clarification on whether the proposed dual cone gradient descent is a completely novel idea, even in the field of multi-objective optimization, or if it is well-known in that field and its application to PINNs is novel. Please clarify and give references if appropriate.
3. Could the authors comment on how their proposed method relates to loss reweighting strategies? It seems that the proposed approach could equivalently be understood as a loss reweighting strategy.
4. Can the authors comment on the case of more than two loss functions.
5. Have the authors considered measuring angles in function space instead of parameter space by using the differential of the parametrization map? Recently, optimization methods that employ function space geometry have shown promise for the optimization of PINNs (see [1, 2, 3]). This might lead to further improvement of the proposed method.

**Details to question 1:**
A fundamental question for me is whether to view PINNs as a multi-objective optimization problem or not. To illustrate this question, consider the following academic example of a PDE with homogeneous forcing and inhomogeneous boundary data:

\begin{align}
Nu &= 0 \quad \text{in }\Omega,
\\
\newline
Bu &= g \quad \text{on }\partial\Omega.
\end{align}

Assume further that $N$ maps constant functions to zero (any PDE operator without a zeroth-order term). Assume a trial neural network is initialized representing (almost) the zero function (or any other constant function). Any change in the neural network will now likely increase the PDE residual (which is at its global minimum for any constant function). Furthermore, the increase in the PDE residual is necessary to obtain the correct solution. This might lead to a Pareto stationary point that has nothing to do with the sought-after solution.

**References**

[1] https://proceedings.mlr.press/v202/muller23b/muller23b.pdf

[2] https://openreview.net/pdf?id=z9SIj-IM7tn

[3] https://arxiv.org/pdf/2402.07318

**Limitations:**

See questions.

---

> ### Author Rebuttal · Authors · 2024-08-06
>
> Thank you very much for your insightful comments and positive feedback.
>
> > **Response to Q1.**
>
> **Simple Answer**: As discussed in Section 3, it is empirically observed that gradient conflicts and imbalances between losses frequently arise during PINN training, which has led to the development of various loss balancing strategies. Our work is motivated by the idea that viewing and addressing these issues from a multi-objective optimization perspective can offer a more effective solution.
>
> **Long Answer**: We appreciate the depth and challenge of this question, as well as the detailed example you provided. We think this question is closely associated with a fundamental aspect of the PINN learning process, which is indeed complex and not yet fully understood both theoretically and empirically.
>
>
> The trade-off between increasing PDE residuals and achieving better approximations to the true solution is indeed a critical question, and there is no theoretical guarantee that this trade-off will always lead to improvements. In the example you mentioned, even if the network is initialized at the trivial solution 0, we think it is possible for the network to reach the true solution smoothly without increasing PDE residuals. This is because, in the absence of boundary conditions, there are infinitely many solutions that satisfy $\mathcal N u=0$, including solutions other than the trivial one. The neural network can transition smoothly among these solutions and approach the true solution.
>
>
> In this context, the main goal of DCGD is to facilitate a smooth learning process without sacrificing PDE residuals. This approach is particularly important in failure modes like the double pendulum example in Section 5.2, where maintaining a smooth and effective learning trajectory is crucial.
>
> >**Response to Q2.**
>
> Thank you for the opportunity to clarify the novelty and originality of our DCGD framework. While the concept of dual cones has been used in convex optimization for duality theory and formulating optimality conditions, our paper employs dual cones to characterize a space where gradient conflicts do not occur and integrates this with gradient descent methods. This approach is novel not only in the context of PINNs but also within gradient descent methods more broadly.
>
>
> For a more detailed discussion of the novelty and originality of DCGD, please refer to "Novelty and Originality of DCGD" in our global response.
>
> >**Response to Q3.**
>
> Thank you for raising this insightful question. While our proposed DCGD framework does share similarities with loss reweighting strategies in that it adaptively adjusts the weight of each loss term (in fact, $\nabla_t \mathcal{L}_{\|\nabla \mathcal{L}_r^{\perp}}$,
>
> $\nabla_t \mathcal{L}_{\|\nabla \mathcal{L}_b^{\perp}}$ in DCGD) at each iteration, it is more accurately described as a gradient surgery approach. This is because DCGD focuses on manipulating the direction of the updated gradient to address gradient conflicts and dominating gradient issues. As discussed in Appendix B, various multi-task learning algorithms [1,2,3] based on gradient surgery techniques can be seen as special cases of DCGD.
>
> Additionally, since loss reweighting schemes operate fundamentally differently, combining DCGD with loss reweighting strategies could offer potential synergies. For example, as shown in Section 5.3, integrating DCGD with existing loss reweighting methods like LRA and NTK leads to improved performance.
>
> [1] Yu, T., Kumar, S., Gupta, A., Levine, S., Hausman, K., & Finn, C. (2020). Gradient surgery for multi-task learning. *Advances in Neural Information Processing Systems*, *33*, 5824-5836.
>
> [2] Liu, B., Liu, X., Jin, X., Stone, P., & Liu, Q. (2021). Conflict-averse gradient descent for multi-task learning. *Advances in Neural Information Processing Systems*, *34*, 18878-18890.
>
> [3 ]Désidéri, J. A. (2012). Multiple-gradient descent algorithm (MGDA) for multiobjective optimization. *Comptes Rendus Mathematique*, *350*(5-6), 313-318.
>
> >**Response to Q4.**
>
> Please refer to “more than two losses” in our global response.
>
> >**Response to Q5.**
>
> Thank you for the interesting suggestion and for bringing the references [4,5,6] to our attention. We have carefully studied these papers and recognize that measuring gradients in function space rather than parameter space offers valuable insights for optimizing PINNs. While more in-depth experimental validation is needed to fully understand gradient conflicts in function space, we have explored this approach by measuring angles using the natural gradient, as illustrated in **Figure 2** of the attached PDF. Our findings indicate that gradient conflicts occur in function space similarly to parameter space. Furthermore, our method appears to handle these conflicts effectively.
> We believe that extending our framework to consider the dual cone in function space could be a promising direction for future research.
>
> [4] Müller, J., & Zeinhofer, M. (2023, July). Achieving high accuracy with PINNs via energy natural gradient descent. In *International Conference on Machine Learning* (pp. 25471-25485). PMLR.
>
> [5] Zeng, Q., Kothari, Y., Bryngelson, S. H., and Schaefer, F. T. Competitive physics informed networks. In The *Eleventh International Conference on Learning Representations.*
>
> [6] Müller, J., & Zeinhofer, M. Position: Optimization in SciML Should Employ the Function Space Geometry. In *Forty-first International Conference on Machine Learning.*

---

> > ### Comment · Reviewer_SdDT · 2024-08-12
> >
> > I thank you very much for the detailed answer.
> >
> > **Concerning Q1:** I understand your argument that the neural network can be trained in a way that does not increase the residual. I am still not fully convinced by the multi-objective viewpoint, but your experimental results are convincing and me not being convinced should not influence my assertion of your work. In any case, it is an interesting question.
> >
> > **Connection to loss re-weighting strategies:**
> > I believe it holds
> > $$
> > G_t \subset \operatorname{span} ( \nabla \mathcal L_b, \nabla \mathcal L_r  )
> > $$
> > hence, no matter which update $g\in G_t$ is picked, we can write
> > $$
> > g = \alpha \nabla \mathcal L_b + \beta \nabla \mathcal L_r
> > $$
> > and hence view the update direction $g$ as stemming from the loss re-weighting
> > $$
> > \tilde{\mathcal L} = \alpha \mathcal L_b + \beta \mathcal L_r
> > $$
> > and $\nabla  \tilde{ \mathcal  L} = g$.
> >
> > **Function spaces**
> > Thank you for the extra experiments. Could you elaborate which metric was chosen to compute the natural gradients?
> >
> > I am convinced that this pre-print merits publication. I will raise my score.

---

> > > ### Author Response · Authors · 2024-08-13
> > >
> > > Thank you for taking the time to review our paper and for raising our score. Below are our responses to your additional questions.
> > >
> > > >**Concerning Q1**
> > >
> > > Thank you once again for your deep and interesting question.
> > >
> > > >**Connection to loss re-weighting strategies**
> > >
> > > We agree with your observation. The proposed DCGD framework can be interpreted as a loss re-weighting strategy that ensures the prevention of gradient conflicts.
> > >
> > > >**Function spaces**
> > >
> > > For computing the natural gradients, we used the metric induced by the inner product in $H^1$ as in [1].
> > >
> > > [1] Müller, J., & Zeinhofer, M. (2023, July). Achieving high accuracy with PINNs via energy natural gradient descent. In *International Conference on Machine Learning* (pp. 25471-25485). PMLR.

---

### Official Review · Reviewer_NFZX · 2024-07-09

**Soundness:** 3
**Presentation:** 2
**Contribution:** 3
**Rating:** 6
**Confidence:** 3

**Summary:**

The paper concerns with the training of physics-informed neural networks (PINNs). It observes that in a multi-objective setting a naive gradient update which decreases the total objective might not decrease every individual objective. This is used to explain the challenge of training PINNs which have a potentially conflicting PDE and a boundary-condition loss. As a remedy, the work proposes to find another descent direction which decreases both of the PINN losses. Three different methods are proposed and the superior "center" method is selected for more detailed empirical investigation. It demonstrates strong results relative to several existing PINN optimizers.

**Strengths:**

The paper presents a clear central story that explains multi-objective optimization in terms of simple geometric concepts. The theoretical analysis in addition to the empirical evaluation is a welcome addition.

**Weaknesses:**

However, the significance of the theoretical contributions must be questioned. There are just two theorems, of which 4.2 is fairly trivial and 4.5 is unclear: why is $\frac{1}{T+1}$ on both sides of Equation 4.2, and what does this equation even state? I cannot make sense of the provided intuition that the convergence rate is $\mathcal{O}(1/T)$, i.e., the more iterations, the slower you converge. Even more so, it is unclear how this intuition follows from Eq. 4.2 in the first place: it says that the sum of gradient norms is upper-bounded by the gap from the initial to the optimal loss, up to some constant.

The work also overlooks some recent literature explaining the difficult behavior of PINNs in terms of poor conditioning and proposing strong optimization strategies, which to the best of my knowledge achieve state-of-the-art convergence. I would argue these ideas should be included in the related work and as baselines contributing to a much more complete story.
- *De Ryck et al. An operator preconditioning perspective on training in physics-informed machine learning. ICLR 2024.*
- *Rathore et al. Challenges in Training PINNs: A Loss Landscape Perspective. ICML 2024.*

Several presentation choices could be improved, including minor mathematical and typing errors, odd wording, and superfluous parts (see suggestions below).

The work also overstates the practical success of PINNs: "a prominent approach for solving PDEs" (line 2), "remarkable empirical performance" (line 4), "PINNs have achieved great success in a wide range of applications" (line 8). This sets a misleading context for the practical significance of PINNs and by extension the presented work.
- *Grossmann et al. Can Physics-Informed Neural Networks beat the Finite Element Method? IMA Journal of Applied Mathematics 2024.*

**Questions:**

- I would kindly ask the authors the clarify Theorem 4.5 in terms of the aforementioned concerns.
- How do the aforementioned optimization strategies compare to your experiments? Do you foresee a way for your method to be compatible to these (similar to section 5.3)?
- Does your method extend to more than two losses? Even in PINNs, there might be additional losses for data and initial conditions.
- How important is the assumption of fixing $\omega_r = \omega_b = 1$? Do the other optimizers show improved performance for other choices of these loss weights?

**Limitations:**

The limitations are discussed rather sparingly and should also include that the method currently applies to just two losses. It should also be noted that the performed experiments are closer to toy than practical problems.

**Errors and suggestions**
- Line 59: It would be helpful if you briefly described what each of these three approaches do, instead of only listing examples.
- Line 73: $d$ is used for both the dimension of the domain as well as the parameter space.
- Eq. 2.2 and line 88: $\omega_u$ should be $\omega_b$
- Line 79: should this be an equality?
- I do not think writing out algorithm 1 adds much value as this is precisely gradient descent but with a different gradient.
- Line 170: should there be an argmin or arginf instead of inf? Or alternatively $\mathcal{L}(\theta*)$ instead of $\theta*$
- Lines 229-240 and Figure 6 might better belong to the appendix as it detracts attention from the main story.
- Line 185: "presents" -> "present"
- Line 221: "ADAM" -> "Adam"
- A.4: the first inequality should be an equality
- Adjust some odd wording, e.g. "mysterious challenges" (line 44), "harmoniously" (line 48)
- I suggest to moderate the claims about the practical success of PINNs (see weaknesses for concrete examples).

---

> ### Author Rebuttal · Authors · 2024-08-07
>
> Thank you very much for your review and for raising questions from the perspective of readers who may be less familiar with optimization theory. We hope the following responses will clarify any misunderstandings and address your concerns.
> > **Response to W1 (Theorem 4.2. is trivial).**
>
> We respectfully disagree with the assessment that Theorem 4.2 is trivial. We believe that this theorem is not only intuitive but also provides significant insights. As detailed in Section 3, much of the existing literature on PINNs has explained gradient pathologies using concepts like gradient conflict and dominating gradients. However, dominating gradients have been somewhat ambiguously defined, often leading to ad hoc strategies. Theorem 4.2 clarifies that dominating gradients should be adaptively defined based on the angle between gradients, addressing a fundamental gap in understanding gradient pathologies that has been overlooked in previous work.
> >**Response to W2 and Q1 (Interpretation of Theorem 4.5).**
>
> Firstly, the statement of Theorem 4.5 (equation 4.5) is standard in optimization theory for showing the convergence of optimization algorithms (for example, see Corollary 2.2.1 of [1] and Theorem 3.2 of [2]).
>
> Additionally, your interpretation of "the more iterations, the slower you converge" is incorrect. The left-hand side of equation 4.5 represents the average of the squared gradient during training, and it implies that:
> $$
> \min_{t=0,1,\ldots, T}\|\nabla\mathcal L(\theta_t)\|^2\leq\frac{1}{T+1}\sum_{t=0}^T\|\nabla\mathcal L(\theta_t)\|^2=\mathcal O(1/T)
> $$
> Thus, the optimization algorithm converges to a stationary point such that $\|\nabla\mathcal L (\theta_t)\|=0$ as the number of iterations increases. This result represents the optimal convergence rate that can be achieved under the condition that only the gradient being Lipschitz continuous is provided.
>
> Furthermore, Theorem 4.5 provides crucial information compared to the convergence results of existing gradient descent methods like SGD and Adam. While SGD and Adam are only guaranteed to convergence to a stationary point, DCGD not only converges to stationary points but also has the potential to converge to Pareto-optimal solutions in multi-objective optimization.
>
> [1] Zhuang, J, et al. Adabelief optimizer: Adapting stepsizes by the belief in observed gradients. NeurIPS 2020.
>
> [2] Hazan, Elad. Lecture notes: Optimization for machine learning. 2019.
> > **Response to W3 and Q1 (Comparison with Two Recent Papers [3,4])**
>
> Thank you for introducing the recent papers. Since one of these papers was published after our submission, we were unable to include it in our related work at that time.  After carefully studying the papers, we compare the motivation, theoretical aspects, and performance of the papers with ours:
>
> **Motivation:**
>
> Both papers focus on the ill-posedness of PINN loss and discuss the importance of preconditioning and the combination of second-order and first-order methods. In contrast, our work analyzes PINN gradient pathology from a multi-objective optimization perspective and proposes a novel algorithm to address these issues.
>
> **Theoretical Aspects:**
>
> We kindly point out that the review's statement that “the convergence results in the two papers are state-of-the-art” is somewhat inappropriate because their convergence results cannot be directly compared to ours. The papers derive convergence results for optimizers by making assumptions about the Hessian (PL condition) and the overparameterized model (assumption that the model can achieve zero loss on the training dataset, see assumption 8.1 of [4]).
>
> In contrast, our results are derived under **the minimal assumption** that gradients are Lipschitz continuous, without any assumptions on the Hessian or the overparameterized model. Thus, the theoretical settings and assumptions are completely different, making a direct comparison of theoretical results inappropriate. To reiterate, our derived convergence rate represents the optimal rate that can be achieved when the Lipschitz continuity condition of the gradient is only provided.
>
> **Performance:**
>
> Our paper and the two mentioned papers take completely different angles on PINN optimization. Thus, combining these ideas harmoniously could lead to further improvements. Indeed, our additional experiments show that the combination of NNCG from [4] with DCGD results in improved performance, see **Table 4** of the attached PDF.
>
> [3] De Ryck et al. An operator preconditioning perspective on training in physics-informed machine learning. ICLR 2024.
>
> [4] Rathore et al. Challenges in Training PINNs: A Loss Landscape Perspective. ICML 2024.
>
> >**Response to W4. overstatement of the practical success of PINNs**
>
> Thank you for pointing this out. We are happy to adjust our language to be more neutral in the revised manuscript.
> >**Response to Q4 ($w_r=w_b=1$)**
>
> The setting $w_r = w_b = 1$ is a common choice in the PINN literature for simplicity in presentation (e.g., [5, 6, 7] ). Furthermore, since DCGD updates using a dual cone region to avoid gradient conflicts across different values of $w_r$ and $w_b$, this ensures robust performance of DCGD regardless of the specific values chosen for $w_r$ and $w_b$. To support this, we have compared the performance of optimizers under different values of weights. As shown in **Table 3**, DCGD consistently demonstrates robust and superior performance compared to other competitors across different values of $(w_r, w_b)$.
>
> [5] Raissi, M., et al. (2019). Physics-informed neural networks: A deep learning framework for solving forward and inverse problems involving nonlinear partial differential equations. Journal of Computational physics.
>
> [6] Wang, S., et al. (2021). Understanding and mitigating gradient flow pathologies in physics-informed neural networks. Journal on Scientific Computing.
>
> [7] Zhao, Z., et al. PINNsFormer: A Transformer-Based Framework For Physics-Informed Neural Networks. ICLR 24.

---

> ### Comment · Reviewer_NFZX · 2024-08-12
>
> I kindly thank the authors for their insightful and strong rebuttal.
>
> I was indeed not aware of this as a standard notation in the optimization literature. I appreciate you pointing this out, including the references, and I acknowledge my oversight on Theorem 4.5. I will also reduce my confidence rating on these grounds.
>
> I also appreciate the theoretical and empirical comparison to the invoked references. For clarification, I referred to [4] achieving the lowest error, not the best convergence rate. It is impressive to see that DGCD can replace Adam + L-BFGS, when fine tuning with NNCG. If possible, could you also report the performance of just DGCD on this task?
>
> I understand that this is combining the DGCD and NNCG sequentially. I assume these can also be combined simultaneously by computing the gradients from NNCG for each loss and then combining them according to your method?
>
> Given that my main concerns have been addressed, as well as the discussion on the broader applicability beyond PINNs and extension to >2 losses, I will adjust my rating.

---

> ### Author Response · Authors · 2024-08-13
>
> Thank you very much for your valuable comments and raising our score.
>
> Here are our responses to your additional questions.
>
> - could you also report the performance of just DGCD on this task?
>
>     The table below shows the performance of DCGD  without fine tuning with NNCG.
>
>     | Optimizer | $L^2$ error |
>     | --- | --- |
>     | Adam | 2.12e-2 |
>     | Adam + L-BFGS | 1.92e-2 |
>     | DCGD | **1.20e-2** |
>
> - I assume these can also be combined simultaneously by computing the gradients from NNCG for each loss and then combining them according to your method?
>
>     There are two methods to combine DCGD with the straetegy of [1].
>
>     **Combination 1**. Apply DCGD and NNCG sequentially
>
>     First, train using DCGD for a certain number of epochs, and then fine-tune with NNCG (as similar to the approach of ADAM + L-BFGS + NNCG).
>
>     **Combination 2**. Apply the updated gradients of DCGD to Each Optimizer
>
>   For each optimizer—ADAM, L-BFGS, and NNCG—the updated gradients from DCGD are used, as described in the pseudo-code provided below.
>
>     ```jsx
>     adam_optimizer = Adam(model_params, adam_params)
>     lbfgs_optimizer = L_BFGS(model_params, lbfgs_params)
>     nncg_optimizer = NNCG(model_params, nncg_params)
>     # for combined: adam_DCGD = True, lbfgs_DCGD = True, nncg_DCGD = True
>     # for sequential (DCGD + NNCG): adam_DCGD = True, lbfgs_epoch = 0, nncg_DCGD = False
>     adam_DCGD = True
>     lbfgs_DCGD = True
>     nncg_DCGD = True
>
>     for i in range(adam_epoch):
>     	if adam_DCGD == True:
>     		# dcgd_closure set the param.grad as the dcgd update vector
>     		adam_optimizer.step(dcgd_closure)
>     	else:
>     		# use loss.backward() where loss = loss_b + loss_r
>     		adam_optimizer.step(closure)
>
>     for i in range(lbfgs_epoch):
>     	if lbfgs_DCGD == True:
>     		# dcgd_closure set the param.grad as the dcgd update vector
>     		lbfgs_optimizer.step(dcgd_closure)
>     	else:
>     		# use loss.backward() where loss = loss_b + loss_r
>     		lbfgs_optimizer.step(closure)
>
>     for i in range(nncg_epoch):
>     	if nncgd_DCGD == True:
>     		# dcgd_closure set the param.grad as the dcgd update vector
>     		nncg_optimizer.step(dcgd_closure)
>     	else:
>     		# use loss.backward() where loss = loss_b + loss_r
>     		nncg_optimizer.step(closure)
>     ```
>
>
> The results shown in Table 4 of the attached PDF were obtained using **Combination 2**. Additionally, we have also experimented with **Combination 1** and achieved further improvements. Due to time constraints, we conducted the experiments under the default parameter settings. Thus, there might be room for further improvement with hyperparameter tuning.
>
> | Optimizer | $L^2$ error |
> | --- | --- |
> | Adam+L-BFGS+NNCG | 9.92e-3 |
> | Combination 1 | **5.60e-3**|
> | Combination 2 | 9.49e-3 |
>
>
> [1] Rathore et al. Challenges in Training PINNs: A Loss Landscape Perspective. ICML 2024.

---

### Official Review · Reviewer_Bm5Z · 2024-07-12

**Soundness:** 3
**Presentation:** 3
**Contribution:** 3
**Rating:** 6
**Confidence:** 4

**Summary:**

In this paper, it is identified that PINNs can be adversely trained when gradients of each loss function exhibit a significant imbalance in their magnitudes and present a negative inner product value. To address these issues, the authors propose a novel optimization framework, Dual Cone Gradient Descent (DCGD), which adjusts the direction of the updated gradient to ensure it falls within a dual cone region. This region is defined as a set of vectors where the inner products with both the gradients of the PDE residual loss and the boundary loss are non-negative. Theoretically, the convergence properties of DCGD algorithms in a non-convex setting are analyzed. On a variety of benchmark equations, DCGD outperforms other optimization algorithms in terms of various evaluation metrics, achieves superior predictive accuracy and enhances the stability of training for failure modes of PINNs and complex PDEs.

**Strengths:**

1. Theoretical analysis is provided, including the necessary and sufficient conditions under which the total gradient falls within the dual cone region, and the convergence rate.
2. Superior performance.

**Weaknesses:**

1. It is better to provide the visualization of PINN predictions and compare with exact solutions, especially for the failure modes of PINNs.
2. Training timing is not reported.

**Questions:**

Line 277, "Failure model" vs. "Failure modes"?

**Limitations:**

The proposed Dual Cone Gradient Descent (DCGD) optimization method for PINNs may still be trapped in local minima.

---

> ### Author Rebuttal · Authors · 2024-08-06
>
> Thank you very much for your positive feedback and suggestions.
>
> - **Response to Weaknesses (W)**
>
> > **W1.Visualization of PINN predictions and exact solutions.**
>
> PINNs predictions and the exact solution, along with absolute error plots, can be found in Appendix C of the original manuscript. Please refer to Figures 9-15 of the original manuscript.
>
> > **W2. Training time**
>
> We provided a comparison of the training speed of the proposed method with other optimization algorithms in Appendix D.2 of the original manuscript. For additional details on the computational cost, please refer to our global response.
>
> - **Response to Questions (Q)**
>
> > **Q1. Line 277**
>
> Thank you for the correction.
>
> - **Response to Limitations**
>
> It is indeed a common challenge in high-dimensional nonconvex optimization problems for all optimization algorithms to potentially get trapped in local minima. However, DCGD has a significant advantage in that it not only converges to stationary points but also to Pareto-stationary points, distinguishing it from methods like SGD and Adam that only guarantee convergence to stationary points. This distinction is a key contribution of our work, as demonstrated in Theorem 4.5. Additionally, the empirical results from the toy example shown in Figure 6 effectively support this theoretical result.
>
> Moreover, integrating noise injection techniques, such as Stochastic Gradient Langevin Dynamics (SGLD) [1], could further enhance the exploration of better local minima, although we have not investigated such variations in this paper.
>
> [1] W. Max and Y. Teh. "Bayesian learning via stochastic gradient Langevin dynamics." ICML, 2011.

---

> > ### Comment · Reviewer_Bm5Z · 2024-08-12
> >
> > Thank the authors for their replies. I am satisfied with their answers and will maintain my score.

---

> > > ### Author Response · Authors · 2024-08-13
> > >
> > > Thank you once again for reviewing our work. We are pleased that you are satisfied with our responses.

---

### Official Review · Reviewer_Nfs2 · 2024-07-14

**Soundness:** 4
**Presentation:** 4
**Contribution:** 4
**Rating:** 8
**Confidence:** 4

**Summary:**

PINNs have become commonplace in both scientific computing and machine learning communities and have found widespread applications. There have been many modifications to various aspects of PINNs, such as the loss functions, initialisations, initial and boundary condition representations, learning algorithms and network architectures.
A major problem with learning effective solutions through PINNs relative to other learning tasks is their difficulty in converging to the actual solution due to the complex nature of the underlying physics and equations involved and the corresponding loss landscape.
The authors have provided a comprehensive survey of the previous works, making the unaddressed problems and goals very clear. A novel algorithmic framework based on convex optimization and geometric analysis is presented, which demonstrates superior performance on many tasks.
The contributions of this paper are multifold, one being a new optimization algorithm while the other being a unified, open and flexible framework to adapt, modify and integrate the current technique with previous and future techniques.
The paper is well written with enough information on the theoretical and mathematical underpinnings, experimental setup, implementation details, results and justifications.

**Strengths:**

Although the concept of dual cone is well known in the optimization domain, specifically in the convex optimization area, the authors utilise and combine existing techniques from optimization theory in a novel way to develop a new algorithmic framework, which they refer to as DCGD.

The authors propose the DCGD algorithm which involves picking a gradient from the Dual Cone region formed by the two losses (boundary loss and residual loss) of PINNs and also theoretically prove that it converges to a pareto-stationary point. Thus, along a new algorithm, theoretical guarantees for convergence are provided, which are very crucial for the goal of improving the training convergence of PINNs.

With a suitable candidate being chosen as the new gradient from an explicitly defined space (called here as $G_t$), multiple variants of the DCGD algorithm can be created. The DCGD can also be combined with many existing optimization techniques, such as Adam Optimizer and Neural Tangent Kernel (NTK).

This work introduces a fairly novel optimization technique which has huge potential for applications in PINNs and multi-objective optimization. The authors have provided strong justification for the generality of their work by showing that most previous MTL approaches are special cases of DCGD. The general, modular and extensible nature of the work paves the way to significantly accelerate further research in various allied areas.

The supplementary material includes the datasets and code for all the experimental details mentioned in the paper and the appendices. I have glanced through the code and data files provided but have not personally verified the results by running the code. (might update here if I get to install and run the code provided)

**Weaknesses:**

The authors have provided enough empirical and theoretical evidence for their proposed DCGD method, though since this work proposes a theoretical and algorithmic framework, albeit it would be notable to have a section on comparing the computational complexity and qualitative terms with existing algorithms. I would say a short section in Appendix D/E would go well.

Since the problem at hand is of multi-objective optimization which overlaps with multi-task learning problems and PINNs are a natural suitable choice, it would be very helpful to see the results of this approach on PINOs (Physics Informed Neural Operators). There is no comparison (even qualitative) of DCGD on PINOs (There are PINOs which are trained only on residual loss, but here I'm specifically referring to PINOs which include both residual and boundary loss for being relevant to DCGD). Was this not considered or omitted, or is it expected to show improvement in performance, as PINNs have shown? (Personally, I believe it should show improvements as the authors have already demonstrated superior performance on PINNformer with DCGD)


Presentations/Grammar/Typos (I do understand that the authors would be well aware of these and would rectify these in the pre-print or camera-ready versions. However, I would like to mention them here together to aid them. Some might have been missed; might update if needed)

Equation 2.2: The weight of the boundary loss is mentioned as $\omega_u$ and also on line 80, whereas it is mentioned as $\omega_b$ (which seems more appropriate) on line 79

Section 4.3: (This is a nitpick and can be conveniently ignored)
$g_t^c$ could be written on the same line as eq. 4.3 or as a newline as eq. 4.4 similar to how equation E.3 is written

Section 5.2: Is the title intended to be “P(I)DEs” or “PDEs”? I believe this is just a typo

Appendix A, line 769: Equations A.8 and A.9 could be mentioned here (as A.4 and A.5) and then referenced in the Proof of Proposition 4.6 section. I believe this will greatly enhance readability

Appendix A: Proof of Theorem 4.2 line 757: The residual loss is repeated twice $\nabla\mathcal{L}_r(\theta_t)$ is repeated twice, it should be both boundary and residual loss, $\nabla\mathcal{L}_r(\theta_t)$ and $\nabla\mathcal{L}_b(\theta_t)$

Appendix A, line770: "One can derive that …" should we include the boundary loss term instead of residual loss as the derivation with residual loss is already shown in line769 i.e it should be $\langle g, \nabla L_b (\theta_t) \rangle \geq 0$

Appendix A, line 781: For the sake of notational consistency, could you replace "condition (2)" with "condition (ii)"

Appendix A, line 785: (nitpick) The subscript t is omitted in the boundary loss and included in the residual loss ($\nabla\mathcal{L}_r(\theta_t)$ and $\nabla\mathcal{L}_b(\theta)$). I suggest the subscript "t" be included at both the places for notational consistency

Appendix A, line 818, Proof of Corollary 4.7:  3. DCGD (Center): “vector” instead of “veoctr”

**Questions:**

I have gone through the previous works mentioned in detail, and does seems like the problem of multi-objective optimization or even optimization, specifically in the case of PINNs, has not been looked at from the perspective of Dual Cones. Could the authors add some more works related to Dual Cone-based techniques to the previous/related works section?

Typos and presentation issues are already covered in the weaknesses section; however, here, I would like to mention a few pointers where I would like to seek clarification. (This is more from a clarification perspective for me individually than a comment on the work, though I believe these clarifications will help the broader community)

Appendix A, line 779, Proof of Theorem 4.5: In the derivation of line779, from the first step to the next, should there be an equality sign in place of the first inequality sign as we are just distributing the dot product over addition on the first term $\nabla \mathcal{L}(y+t(x-y))$? (i.e. the first of the 3 inequality signs should be an equality instead)

Appendix A line 797: Is the expression $\left| \cos(\phi_t) - \pi \right| < \alpha$ is correct and as intended? The inequalities on lines 810 and 811 and the threshold criterion in the algorithms in Appendix E ($\pi - \alpha < \phi_t \leq \pi$) point otherwise. I believe there should not be the cosine of $\phi$; rather, it should just be the angle/phase difference $\phi$.
I have a similar comment for line 984, as the same expression is used.

Appendix C.5 line 925: Is the statement complete? Is it intentionally left this way?

**Limitations:**

I am appreciative of the authors who have mentioned (and also acknowledged in the checklist) a few limitations of the work in section 6 this is more from a future work perspective than from a limitations of the work presented as part of this paper. The limitations of the current work are not very evident from the present text.

It would be further helpful if they could include a more detailed section on where this cannot be applied or other multi-objective scenarios where the DCGD optimization framework is not as effective as other previous techniques.

---

> ### Author Rebuttal · Authors · 2024-08-06
>
> Thank you for your  insightful comments, and for thoroughly checking details such as typos. We deeply appreciate your high-quality review.
>
> - **Response to Weaknesses (W)**
>
> > **W1. Comparison of Computational Complexity**
>
> Please refer to the computational complexity discussion in our global response.
>
> > **W2. Application to PINOs**
>
> Thank you for your valuable suggestion regarding the application of DCGD to PINOs. We have conducted additional experiments on PINOs for Burgers' equation and Darcy flow. The results are summarized in **Table 1** of the attached PDF, which confirm that DCGD can also enhance the performance of PINOs. For further details on the broader implications and potential impact beyond PINNs, please refer to our global response.
>
> > **W3. Typos**
>
> Thank you for carefully checking the details. We will incorporate the suggested corrections into the revised manuscript. One point to clarify: “P(I)DEs” refers to partial integro-differential equations.
>
> - **Response to Questions (Q)**
>
> > **Q1. Previous Work Related to Dual Cone-Based Optimization**
>
> Dual cones are a fundamental concept in optimization, particularly in convex optimization. They are used to characterize optimal solutions and feasibility conditions in constrained convex optimization and duality theory. However, to the best of our knowledge, there have been no previous instances of incorporating dual cones into gradient manipulation or developing a dual cone-based gradient descent method. We emphasize that our proposed framework is not merely an application of existing methods to multi-objective problems or PINNs but represents a completely novel development in the field. For further discussion on the novelty and originality of our work, please refer to our global response.
>
> > **Q2. Derivation of line 779**
>
> You are correct. We perform the distributive property of the inner product, which results  in an equality. Thus, the inequality still (obviously) holds, but for clarity, we will change it to an equality as suggested.
>
> > **Q3. Line 797**
>
> Thank you for your comment. The threshold $\alpha$ is set to stop the algorithm when it reaches near the Pareto-stationary point. Therefore, the cosine should be removed from the equation, i.e., the expression $\vert \cos(\phi_t) - \pi \vert < \alpha$ should be changed to $\pi - \alpha < \phi_t \leq \pi$ .
>
> > **Q4. Line 925.**
>
> Thank you again for your careful review. The sentence that should have been removed during the annotation process was accidentally included.
>
> - **Response to limitations**
>
> While this paper demonstrates the valuable result that DCGD converges to Pareto-stationary points, which is a notable advantage over popular optimizers like SGD and Adam, there are still unexplored aspects regarding its convergence. For instance, an important and challenging question is whether DCGD can be guided to converge to a better Pareto-optimal point within the Pareto set. Additionally, although we proposed three algorithms—Center, Projection, and Average—within the DCGD framework and found DCGD (Center) to be the most effective for PINNs, there may be more efficient algorithms for other problems (e.g., PINO, MTL). These possibilities were not addressed in this paper and represent interesting directions for future research.

---

> ### Comment · Reviewer_Nfs2 · 2024-08-10
>
> I have read through all the reviews, rebuttals and responses (including the general author rebuttal). I am satisfied with the authors' response and the general comment. I am pleased to see the applications of DCGD in unlearning problems. I would like to retain my review and the associated scores.

---

> > ### Author Response · Authors · 2024-08-11
> >
> > Thank you for your feedback. We are glad that our rebuttals have addressed your concerns.

---

### Author Rebuttal · Authors · 2024-08-06

**Global response to all reviewers**

We would like to express our sincere appreciation for the insightful comments and valuable suggestions provided by all reviewers. We assure you that all comments and suggestions will be thoroughly addressed in the revised manuscript. In this global response, we will summarize and address the common questions and key comments raised by the reviewers.
>  **Novelty and originality of DCGD** (*in response to Reviewers Nfs2, SdDT*)

To the best of our knowledge, our work represents the first instance of applying the concept of dual cones specifically to gradient manipulation within the context of gradient descent. This proposed framework is novel not only for PINNs but also across the broader field of multi-objective optimization.

While dual cones are indeed a fundamental concept in optimization, particularly in relation to constrained convex optimization (e.g., duality theory and formulating optimality conditions for convex problems), as noted by Reviewer Nfs2, our approach introduces a novel application of dual cones to manage gradient conflicts. This innovative use in the context of gradient descent is an area that has not been previously explored.

We would like to highlight the key contributions of our work. Existing strategies to resolve gradient conflicts are often ad hoc and lack a systematic framework. In contrast, our method provides a principled approach by utilizing the dual cone characterization to avoid gradient conflicts, as detailed in Theorem 4.2 and Proposition 4.3. This approach allows DCGD to be viewed as a generalization of various multi-task learning (MTL) algorithms, as discussed in Appendix B. Consequently, the proposed method has the potential to advance the development of future multi-objective optimization algorithms.Furthermore, our convergence results, presented in Theorem 4.5, demonstrate that DCGD converges not only to stationary points but also to Pareto-stationary points in a multi-objective non-convex setting. This provides both theoretical and empirical advantages compared to methods such as SGD and Adam, which guarantee convergence only to stationary points.

> **Broader Implications and Potential Impact Beyond PINNs** (*in response to Reviewer Nfs2*)

Our initial focus was on PINNs, but as suggested by Reviewer Nfs2, DCGD can be applied to various modern machine learning problems where multiple loss functions need to be managed simultaneously, such as in Physics-Informed Neural Operators (PINO), multi-task learning (MTL), and machine unlearning (where both forgetting and retraining losses must be considered).

To demonstrate the extensibility and applicability of DCGD, we have conducted additional experiments applying DCGD to PINO (in response to reviewer Nfs2) and machine unlearning problems. As shown in **Table 1** in the attached PDF, DCGD improves performance compared to optimally tuned methods when applied to PINO. Furthermore, in the context of machine unlearning, DCGD enables unlearning to be performed without compromising the quality of the generated images, as illustrated in **Table 2** and **Figure 1** in the attached PDF file.

> **More than two losses** (*in response to Reviewers NFZX, SdDT*)

- **Easy and simple approach**: For cases involving more than two loss functions, such as those arising from multiple governing equations and boundary conditions in PINNs, DCGD can be effectively applied by treating the multiple losses as a combination of residual loss and the other losses. For instance, in Section 5.2, we applied this approach to A-PINNs with three loss functions and obtained improved results.
- **More general approach**: Let us consider the total loss function $\mathcal{L}(\theta)$, which is the sum of multiple loss terms:
$$ \mathcal{L}(\theta):= \sum^{n}_{i=1}\mathcal{L}_i (\theta),  \text{where }i=1,2, \cdots, n $$

Denote by $\mathbf{K}^*_t$ the set of vectors satisfying $\langle u, \nabla \mathcal{L}_i(\theta_t) \rangle \geq 0$ for all $i = 1, 2, \cdots, n$. Then we can characterize $\mathbf{G}_t$, which is a subset of $\mathbf{K}^*_t$, by similar approach developed in this paper. For example, let $\mathbf{G}^{ij}_t$ be a subset of a dual cone region for $\nabla \mathcal{L}_i(\theta)$, $\nabla \mathcal{L}_j(\theta)$ which defined as follow:



$$\mathbf{G}_t^{ij} := \big\lbrace c_1g_i+c_2g_j+c_3g_c^{ij} \big| c_1, c_2, c_3\geq 0  \big\rbrace$$

where
$g_k = \nabla_t \mathcal{L}_{\|\nabla \mathcal{L}_k^\perp} $ for $k= i,j$ and $g_c^{ij}$ is a unit vector that is orthogonal to both vectors $g_i, g_j$.

We can get $g_c^{ij}$ by a cross product of  $g_i$ and $g_j$ (More specifically, after express each vector by sum of $\nabla \mathcal{L}_i (\theta), \nabla \mathcal{L}_j (\theta)$ ).

Then we can determine the subset of dual cone as follows: $\mathbf{G_t} := \mathop{\bigcap}\limits_{i\neq j}^{n} \mathbf{G}_t^{ij}$.  In other words, the principles developed in this paper allow for the characterization of the dual cone region even when dealing with more than two loss functions.

> **Computational cost** (*in response to Reviewers Nfs2, Bm5Z*)

We compared the training speed of the proposed method with other optimization algorithms in Appendix D.2 of the original manuscript. It is important to highlight that DCGD algorithms are essentially first-order (explicit) gradient methods. Therefore, while there may be differences in speed due to the complexity of the update rule, these differences in training speed are not significant.

---

### Decision · Program_Chairs · 2024-09-25

**Decision:**

Accept (poster)

**Comment:**

This work introduces a novel approach for solving multi-objective optimization as arises in the training of physics-informed neural networks. The idea is simple and natural: to select an update direction that is guaranteed to be a descent direction for each component of the loss function individually. The authors achieve significant improvement in the resulting accuracy, on a wide range of PINN benchmarks. The approach is nonintrusive in terms of the other aspects of PINN training and might be equally useful for multiobjective training in other domains.

In summary, the method is well-motivated, has potential for broad applicability, and is thoroughly evaluated (at least in the domain of PINNs, which is the focus of this work) I therefore recommend acceptance.